# Set Valued Predictions For Robust Domain Generalization

Ron Tsibulsky [1]   Daniel Nevo [2]   Uri Shalit [2,3]

## Abstract

Despite the impressive advancements in modern machine learning, achieving robustness in Domain Generalization (DG) tasks remains a significant challenge. In DG, models are expected to perform well on samples from unseen test distributions (also called domains), by learning from multiple related training distributions. Most existing approaches to this problem rely on single-valued predictions, which inherently limit their robustness. We argue that set-valued predictors could be leveraged to enhance robustness across unseen domains, while also taking into account that these sets should be as small as possible. We introduce a theoretical framework defining successful set prediction in the DG setting, focusing on meeting a predefined performance criterion across as many domains as possible, and provide theoretical insights into the conditions under which such domain generalization is achievable. We further propose a practical optimization method compatible with modern learning architectures, that balances robust performance on unseen domains with small prediction set sizes. We evaluate our approach on several real-world datasets from the WILDS benchmark, demonstrating its potential as a promising direction for robust domain generalization.

## 1. Introduction

In recent years, Machine Learning (ML) methods, particularly deep neural networks (DNNs), have achieved remarkable success in various tasks, including language understanding and image recognition. However, despite these advancements, many models struggle in real-world scenarios when the data comes from a different distribution than the data the model was trained on (Nagarajan et al., 2020; Miller et al., 2020; Recht et al., 2019). These distributions are referred to as "domains", and the challenge of generalizing to domains unseen during training time is known as Out-Of-Distribution (OOD) generalization, or Domain Generalization (DG) (Li et al., 2018a; Krueger et al., 2021). DG is especially important in high-stakes fields like medicine, where, for instance, a disease diagnosis system might be trained on data from a specific group of patients, each with records from multiple visits to health centers, or multiple medical scans. Such a system must be effective on new patients for which it has no prior data. Moreover, beyond merely performing well on average, it is desirable that such a system provide robust performance across all new patients, requiring worst-case performance guarantees across domains rather than average-case performance.

Recent efforts to address this challenge have focused on learning "stable" predictors that focus only on domain-invariant features, discarding information that might be domain-specific (Peters et al., 2016; Arjovsky et al., 2019; Heinze-Deml & Meinshausen, 2021; Veitch et al., 2021; Wald et al., 2021). While these methods have made strides in maintaining stability across various domains, achieving this stability remains challenging, and sometimes infeasible (Chen et al., 2021; Kamath et al., 2021). Furthermore, stability often comes at the cost of sub-optimal and even poorly-performing predictors, as "overly stable" predictors might ignore valuable parts of the data (Rosenfeld et al., 2021; 2022; Zhao et al., 2019; Stojanov et al., 2021). As an illustration, a disease diagnosis system that consistently achieves only 60% accuracy, despite being stable across different patient populations, may not be practically viable.

We propose an alternative approach for dealing with the considerable challenge of DG: employing predictors with *set-valued outputs*. Our set-valued framework for OOD problems recognizes that a key challenge in distribution-shift problems is the absence of a single, universally optimal prediction across all domains, as different domains induce different probabilistic relationships between features $X$ and label $Y$. To handle these variations, our approach provides set-valued outputs that capture a range of potential relationships learned from the training domains. Such an approach is sensible for multi-domain settings, where expecting a single predictor to perform optimally across all domains is

---

[1]Department of Computer Science, Tel Aviv University, Tel Aviv, Israel [2]Department of Statistics and Operations Research, Tel Aviv University, Tel Aviv, Israel [3]Department of Data and Decisions Science, Technion, Haifa, Israel. Correspondence to: Ron Tsibulsky <ront65@gmail.com>.

*Proceedings of the 42nd International Conference on Machine Learning*, Vancouver, Canada. PMLR 267, 2025. Copyright 2025 by the author(s).

often impractical, yet worst-case performance guarantees are still desirable.

A useful set-valued prediction should not only include the correct label with high probability, it should ideally output small prediction sets. Consider a degenerate predictor that always predicts the entire label space $\mathcal{Y}$. While it would always include the correct label, such a predictor will have no practical value. Therefore, it is essential to aim for a predictor that achieves a desirable performance level, while keeping the prediction set size to a minimum. This observation represents our learning objective: instead of seeking to maximize performance with a singleton set, we suggest targeting a predefined performance level while minimizing the prediction set size.

To summarize, our contributions in the paper are as follows: In **Section 2** we introduce a learning paradigm for DG problems with set-valued predictors, where the goal is to achieve a predefined coverage level across unseen domains. In **Section 3** we formulate and prove generalization bounds with respect to the above paradigm, giving conditions which guarantee robust performance on unseen domains.
In **Section 4** we propose practical methods for learning set-valued predictors within the above paradigm.
Finally, in **Section 5** we demonstrate the effectiveness of our proposed methods using real-world datasets from the WILDS benchmark (Koh et al., 2021).

## 2. Multi Domain Set Valued Learning

### 2.1. Problem Setting

Consider features $X \in \mathcal{X}$, a label $Y \in \mathcal{Y}$, and a domain $e \in \mathcal{E}$ coupled with a distribution $P_e$ over $(\mathcal{X}, \mathcal{Y})$. In this work, we focus on classification tasks, meaning $\mathcal{Y}$ is a known finite set. Let also $D$ be a distribution over the set of domains $\mathcal{E}$. This setting can be viewed as a meta-learning problem, where the data generation process is hierarchical, involving distributions at both the domain and instance levels.

We assume that during training we observe a set $\mathcal{E}_{train}$ of $m$ domains drawn according to $D$. From each domain $e \in \mathcal{E}_{train}$, we observe a sample $S_e$ of $n_e$ instances, sampled according to $P_e$. This brings us to an overall sample $S = \bigcup_{e \in \mathcal{E}_{train}} S_e$ of $N = \sum_{i=1}^{m} n_e$ instances of $(X_i, Y_i, e_i)$. At test time, we observe an instance $X$ sampled from some domain $e_{test}$ (sampled according to $D$), which may differ from any of the training domains. We note that while we observe the value of $e$ during training, it is neither required nor utilized at test time.

Throughout this paper we consider set-valued predictors. Such predictors take inputs from $\mathcal{X}$, and output a subset of $\mathcal{Y}$, i.e $h : \mathcal{X} \longrightarrow \mathcal{P}(\mathcal{Y})$.

### 2.2. Learning Objective

To formalize our learning objective, we first define a per-domain loss function $\mathcal{L}(h, e)$, which measures the performance of a set-valued predictor $h$ within a specific domain $e$. For set-valued predictors, the loss function should evaluate the *coverage* of the prediction set. Since set-valued predictors can be decomposed into per-label binary classifiers, it is natural to require that each of these classifiers performs well, especially when worst-case performance is desirable. For instance, in disease diagnosis it is crucial to prioritize per-label recall for each $y \in \mathcal{Y}$, to avoid missed diagnoses. The following loss function captures this focus on per-label recall:

$$\mathcal{L}_{recall}(h, e) = \max_{y \in \mathcal{Y}} P_e[y \notin h(X)|Y = y].$$

Balancing this loss with the size of the prediction set ensures that set-valued predictors include each correct label $Y$ when appropriate while refraining from including incorrect ones, ensuring that the prediction sets are both informative and concise. This balance mirrors the trade-off between recall and precision in single valued predictors.
The focus on recall loss for set-valued predictors is also adapted by other works on set-valued predictors, such as Mauricio Sadinle & Wasserman (2019); Wang & Qiao (2018); Guan & Tibshirani (2022).

Returning to the disease prediction example, when developing a model to predict diseases using data from multiple patients each with multiple visits, it is desirable that the model performs robustly on new patients, rather than merely perform well on average. To obtain this, we begin by defining a performance indicator for a loss function $\mathcal{L}$ and a threshold $\gamma$ as: $\mathbb{1}_{\mathcal{L},\gamma}(h, e) = \mathbb{1}[\mathcal{L}(h, e) \geq \gamma]$. Our objective then becomes:

$$\min_{h \in \mathcal{H}} \mathbb{E}_{e \sim D} \mathbb{1}_{\mathcal{L},\gamma}(h, e) = \min_{h \in \mathcal{H}} P_{e \sim D}[\mathcal{L}(h, e) \geq \gamma].$$

Achieving a zero objective implies that our predictor delivers a satisfying performance (according to $\mathcal{L}, \gamma$) across all domains. For example, with $\mathcal{L}_{recall}$ loss, achieving an objective value of zero would mean our predictor achieves at least $1 - \gamma$ recall for all labels in $\mathcal{Y}$, across all domains. In practice, we aim for a low objective value, though not necessarily zero. With generalization bounds on

$$\left| \mathbb{E}_{e \sim D} \mathbb{1}_{\mathcal{L},\gamma}(h, e) - \frac{1}{N} \sum_{e \in \mathcal{E}_{train}} \mathbb{1}_{\mathcal{L},\gamma}(h, e) \right|,$$

we can bound the expected fraction of domains where a predictor does not deliver satisfying performance. A significant advantage of this approach is that it frames the robustness problem as a mean optimization problem, allowing to leverage established methods from classical ML and adapt them to the multi-domain context. An example using the concept of VC dimension is presented in Section 3.

## 2.3. Related Work

Set-valued predictions are well studied in the single-domain framework, often through the lens of conformal prediction - a widely used method for constructing set-valued predictions with a predefined level of coverage guarantee (Shafer & Vovk, 2008; Vovk et al., 2005); see also Section 4.2.

In the single-domain scenario, Mauricio Sadinle & Wasserman (2019) and Lei (2014) tackle the challenge of aiming at set-valued predictions with a pre-defined level of coverage, while keeping the prediction sets small. They present that challenge as a constrained optimization problem and define its solution using acceptance regions, leveraging Neyman-Pearson Lemma to construct optimal regions.

Wang & Qiao (2018; 2023a) tackle the same constrained optimization problem using support vector machines, building another method for set-valued predictions in the single-domain scenario.

Dunn et al. (2018) suggest set-valued predictions for DG tasks, however they focus on optimizing average risk over domains, instead of worst-case type risk as we do in our work. In addition, the work of Dunn et al. (2018) does not consider the size of the prediction set, and does not aim at small prediction sets.

Wang & Qiao (2023b) also use set-valued predictions in OOD setting, continuing the work on support vector machines for optimal set-valued predictions. However, they assume the conditional distribution $P_e[X|Y]$ is constant across domains, and specifically tackle OOD detection of unseen classes, when unlabeled data from the test domain is available. For this setting, they derive an empirical optimization problem similar to the one we derive for our setting, which is presented in section 4.1.

Guan & Tibshirani (2022) tackle an OOD problem with a single train domain and a single test domain, where unlabeled data from the test domain is available, and use conformal prediction to achieve per-label recall guarantee.

One work that approaches worst-case robustness in DG tasks with a single-valued predictor is that of Eastwood et al. (2022) which suggests the Quantile Risk Minimization (QRM) method for minimizing the $\alpha$-quantile risk among domains. This method is built upon an assumption that there exists a high-level distribution over the domains, same as we do in our work. However, the applicability of this method is constrained by its reliance on single-valued predictions, which, as discussed in Section 1, can lead to suboptimal and degraded performance in certain domains.

## 3. Generalization Bounds

In the previous section we introduced the concept of a performance indicator $\mathbb{1}_{\mathcal{L},\gamma}(h,e)$ measuring whether the predictor $h$ gives satisfactory performance in domain $e$ (w.r.t. $\mathcal{L},\gamma$), and set forth the objective of

minimizing $\mathbb{E}[\mathbb{1}_{\mathcal{L},\gamma}(h,e)]$. We argued that bounding $|\mathbb{E}_{e\sim D}\,\mathbb{1}_{\mathcal{L},\gamma}(h,e) - \frac{1}{N}\sum_{e\in\mathcal{E}_{train}}\mathbb{1}_{\mathcal{L},\gamma}(h,e)|$ will allow us to focus on minimizing the empirical average of the performance indicator over training domains.

This concept mirrors the way Probably Approximately Accurate (PAC) bounds provide a foundation for Empirical Risk Minimization (ERM) in classical ML (Shalev-Shwartz & Ben-David, 2014). We therefore explore how classical machine learning tools can be adapted to the hierarchical structure of domains. In this section, we will formally define the concept of VC-dimension within our hierarchical framework and derive generalization bounds based on this definition. While definitions and generalization bounds adapt naturally from classical ML to our setting, we show that in the multi-domain set-valued context, even basic hypothesis sets like linear hypotheses, which have a finite VC-dimension in standard supervised learning, have infinite VC-dimension when no assumptions on $\mathcal{E}$ are made. Nevertheless, we also demonstrate that under certain assumptions on $\mathcal{E}$, such hypothesis sets retain finite VC-dimension.

### 3.1. VC-dim In Meta Learning - Definitions and Generalization Bounds

One of the classical results in the context of standard ML considers generalization of binary classifiers with regard to 0-1 loss $\mathcal{L}_{0-1}(h,(x,y))$. The $\mathcal{L}_{0-1}$ loss serves as a measure for whether an hypothesis $h$ performs well on a data point $(x,y)$. The VC-dimension quantifies the ability of an hypothesis set $\mathcal{H}$ to exactly classify any subset of points from a finite set in $\mathcal{X}\times\mathcal{Y}$. To adapt the idea of VC-dimension, and its implications on generalization, we start by replacing the $\mathcal{L}_{0-1}$ loss with the performance indicator $\mathbb{1}_{\mathcal{L},\gamma}(h,e)$ as the measure of how well an hypothesis $h$ performs on a domain $e$. Then, we define the VC-dimension such that it measures the ability of an hypothesis set $\mathcal{H}$ to perform perfectly on any subset of domains from a finite set in $\mathcal{E}$.

**Definition 3.1** (Shattering). Given a hypotheses set $\mathcal{H}$, a finite set $C\subseteq\mathcal{E}$, a loss function $\mathcal{L}$, and a threshold $\gamma$, we say that $\mathcal{H}$ shatters $C$ with respect to $\mathcal{L},\gamma$ if for any subset $\hat{C}\subseteq C$ there exists $h\in\mathcal{H}$ such that

$$e\in\hat{C}\longrightarrow\mathbb{1}_{\mathcal{L},\gamma}(h,e)=1 \quad e\in C\backslash\hat{C}\longrightarrow\mathbb{1}_{\mathcal{L},\gamma}(h,e)=0.$$

**Definition 3.2** (VC-Dimension). The VC-dimension of a hypothesis class $\mathcal{H}$ with respect to $\mathcal{L},\gamma$ is the maximum size $M$ such that there exists a set $C\subset\mathcal{E}$ with $|C|=M$ that $\mathcal{H}$ can shatter with respect to $\mathcal{L},\gamma$. We denote it by $VCdim_{\mathcal{L},\gamma}(\mathcal{H})$.

**Definition 3.3** (Uniform Convergence). We say that a hypothesis class $\mathcal{H}$ has the uniform convergence property with respect to loss function $\mathcal{L}$ and threshold $\gamma$ if, for every $\delta,\epsilon\in(0,1)$, there exists an integer $m(\delta,\epsilon)$ such that

for every distribution $D$ over $\mathcal{E}$, if $S \subset \mathcal{E}$ is a sample of $|S| > m(\delta, \epsilon)$ domains drawn i.i.d according to $D$, then, with probability at least $1 - \delta$ (over the sample of $S$) the following holds:

$$\forall h \in \mathcal{H}, \left| \mathop{\mathbb{E}}_{e \sim D}[\mathbb{1}_{\mathcal{L},\gamma}(h, e)] - \frac{1}{|S|} \sum_{e \in S} \mathbb{1}_{\mathcal{L},\gamma}(h, e) \right| \leq \epsilon.$$

**Theorem 3.4.** *A hypothesis class $\mathcal{H}$ has the uniform convergence property with respect to a loss function $\mathcal{L}$ and threshold $\gamma$ if and only if $VCdim_{\mathcal{L},\gamma}(\mathcal{H}) < \infty$. Moreover, if $d = VCdim_{\mathcal{L},\gamma}(\mathcal{H}) < \infty$, the sample size from the uniform-convergence definition is $m(\delta, \epsilon) = \Theta(\frac{d + log(1/\delta)}{\epsilon^2})$.*

Theorem 3.4 is proved in appendix A.1, and provides a way to derive generalization bounds for the hierarchical setting by demonstrating a finite VC-dimension for a hypothesis class $\mathcal{H}$, analogous to how the VC-dimension is used in classical machine learning. While Theorem 3.4 closely mirrors classical VC-dimension theory, its proof requires special care due to two key differences from the standard setting. First, instead of using the 0-1 loss on individual data points, we work with a performance indicator $\mathbb{1}_{L,\gamma}(h, e)$ that measures whether a hypothesis achieves satisfactory performance on an entire domain $e$. Second, classical results apply to distributions over input-label pairs $\mathcal{X} \times \{0, 1\}$, whereas our setting involves distributions over domains $\mathcal{E}$, which are more complicated objects. These differences necessitate a careful adaptation of the uniform convergence argument, including a mapping to a derived hypothesis class over $\mathcal{E} \times \{0, 1\}$ to apply classical VC theory. Further details are outlined in appendix A.1.

Proving a finite VC-dimension for certain hypothesis sets in the hierarchical context with the $\mathcal{L}_{recall}$ loss requires assumptions on the structure of $\mathcal{E}$, as we show in sections 3.2 and 3.3. This is unlike standard results in the classic framework, where assumptions on the inputs are usually not required. Informally, this difference can be seen as stemming from the fact that the space of all domains is of infinite dimension, while in the classical framework inputs are typically considered to belong to finite-dimensional space. In accordance with that, the restrictions on $\mathcal{E}$ that we explore in section 3.3 limit the domains to a finite-dimensional subspace.

In this section, we focus on generalization from training domains to new, unseen domains. We do not address generalization within individual domains, which can be viewed as effectively assuming an infinite sample size for each domain. In appendix B, we provide a comprehensive analysis of both the number of domains required for effective generalization and the sample size needed within each domain.

## 3.2. Failure To shatter By Linear Hypotheses Set

In the following sections, we will consider hypothesis sets of set-valued predictors. These predictors can be defined using $|\mathcal{Y}|$ binary classifiers $h_y : \mathcal{X} \longrightarrow \{0, 1\}$ for $y \in \mathcal{Y}$ in the following manner: $h(X) = \{y \in \mathcal{Y} : h_y(X) = 1\}$. If each $h_y$ adheres to a particular structure (e.g., $h_y$ is a linear classifier), we say that $h$ possesses this structure.

We demonstrate the divergence between classic supervised learning and multi-domain set-valued learning using the example of linear hypotheses sets. In the classical sense of shattering, a linear hypotheses set cannot shatter more than $d+1$ points (Shalev-Shwartz & Ben-David, 2014), thus having a finite VC dimension. However, in the multi-domain set-valued setting, without imposing structural assumptions on the domains in $\mathcal{E}$, the finiteness of the VC dimension depends on the dimensionality $d$.

**Theorem 3.5.** *For $d = 1$, linear $\mathcal{H}$ cannot shatter more than $2|\mathcal{Y}|$ domains with respect to $\mathcal{L}_{recall}$ and any $0 < \gamma < 1$.*

According to Definition 3.2 this means that for linear $\mathcal{H}$ and $d = 1$, we have $VCdim_{\mathcal{L}_{recall},\gamma}(\mathcal{H}) \leq 2|\mathcal{Y}|$.
Proof for theorem 3.5 is given in appendix A.2.

**Theorem 3.6.** *For $d > 1$, and any size $m$, there exists a set of $m$ domains that can be shattered by linear hypotheses $\mathcal{H}$ with respect to $\mathcal{L}_{recall}$ and any $0 < \gamma < 1$.*

According to Definition 3.2 this means that for linear $\mathcal{H}$ and $d > 1$, we have $VCdim_{\mathcal{L}_{recall},\gamma}(\mathcal{H}) = \infty$.
Proof for theorem 3.6 is given in appendix A.3.

## 3.3. Imposing Restrictions on $\mathcal{E}$

The previous results may seem discouraging, as an infinite VC-dimension suggests that generalization from observed to new domains via uniform convergence is impossible. However, when real-world data is collected from various environments, it is reasonable to assume that the Data Generation Processes (DGP) of these environments share some common structure. Thus, we examine cases where restrictions are placed on $\mathcal{E}$. Following Wald et al. (2021); Rosenfeld et al. (2022) we examine the assumption that all domains in $\mathcal{E}$ are conditionally Gaussian:

$$\forall e \in \mathcal{E} \ \forall y \in \mathcal{Y} \ [X|Y = y] \overset{P_e}{\sim} N(\mu_{e,y}, \Sigma_{e,y}).$$

While the previous subsection shows negative results for learnability when there are no restrictions on the domains, the following result suggests that with some restrictions as mentioned above, learnability is possible with sufficient training domains.

**Theorem 3.7.** *If domains are restricted to being Conditionally Gaussian, and there exist $\forall_{y \in \mathcal{Y}} \Sigma_y \in \mathbb{R}^{d \times d}$ such that for each $e, y$ $\Sigma_{e,y} = \sigma_{e,y}\Sigma_y$ for some $\sigma_{e,y} \in \mathbb{R}$, then for*

*each $\epsilon, \delta > 0$ there exist $m = \Theta(\frac{|\mathcal{Y}|^2(d+log(|\mathcal{Y}|/\delta))}{\epsilon^2})$ such that if $|\mathcal{E}_{train}| \geq m$ then with probability higher than $1 - \delta$ over the training domains sample $\mathcal{E}_{train}$, and regardless of the distribution $D$ over $\mathcal{E}$ , for any linear hypothesis $h$*

$$\forall_{e \in \mathcal{E}_{train}} \mathbb{1}_{\mathcal{L}_{recall}, \gamma}(h, e) = 0 \longrightarrow \underset{e \sim D}{\mathbb{E}}[\mathbb{1}_{\mathcal{L}_{recall}, \gamma}(h, e)] \leq \epsilon.$$

The proof is provided in Appendix A.4, building on Theorem 3.4.

Theorem 3.7 provides a rigorous justification for seeking a linear predictor $h \in \mathcal{H}$ that achieves $\mathcal{L}_{recall}(h, e) < \gamma$ for each $e \in \mathcal{E}_{train}$. In the next section, we will introduce practical methods to find such predictors $h$, while also considering the efficiency of the set sizes generated by $h$.

While the result of 3.7 is strictly valid for linear predictors, we experimentally evaluate its applicability to neural networks.

In addition, a limitation of Theorem 3.7 lies in its assumption that for each label $y \in \mathcal{Y}$, the Gaussian DGPs across all domains share the same covariance structure, up to a scaling factor $\sigma_{e,y}$. In Appendix E.3, we empirically demonstrate that generalization is achievable even without this assumption. Specifically, we report the results of experiments on synthetic data with Gaussian DGPs that feature separate random covariance matrix for each domain. Additionally, in Section 5.2 we empirically evaluate the generalization capability of our proposed method (introduced in the next section) to new domains with complex DGPs using real-world datasets.

# 4. Practical Optimization Methods

In previous sections we discussed the necessity of finding a set-predictor $h$ that consistently delivers satisfactory performance across all training domains, as measured by a loss function $\mathcal{L}(h, e)$, while also minimizing the size of the prediction set. In this section, we introduce our method, **SET Coverage Optimized with Empirical Robustness (SET-COVER)**, which addresses this dual objective. In addition, we present a series of baseline predictors that leverage conformal prediction for DG.

To ground our discussion, we will focus on the per-label recall metric as our performance criterion, continuing the earlier discussion on disease diagnosis system. However, the methods we develop can be naturally extended to other performance metrics. Throughout the next sections we will call this metric "min-recall", since achieving a required recall level for all labels simultaneously implies that the minimum recall across labels meets the required threshold.

We begin by introducing some key notations. Given a training dataset $S = \bigcup_{e \in \mathcal{E}_{train}} S_e$, we define: $G_{e,y} =$

$\{i \in S_e : Y_i = y\}$, and $G_y = \{i \in S : Y_i = y\} = \bigcup_{e \in \mathcal{E}_{train}} G_{e,y}.$

## 4.1. SET-COVER: Optimized Set Prediction

Our goal is to minimize prediction set size while maintaining the required min-recall level. We formalize the following constrained optimization problem:

$$\min_{h \in \mathcal{H}} \frac{1}{N} \sum_{i \in S} |h(x_i)|$$

$$\text{s.t.} \frac{1}{|G_{e,y}|} \sum_{i \in G_{e,y}} \mathbb{1}[y \notin h(X_i)] \leq \gamma, \ \forall e \in \mathcal{E}_{train} \ \forall y \in \mathcal{Y}.$$

Since $|h(X)| = \sum_{y \in \mathcal{Y}} h_y(X)$, this problem can be solved for each label $y$ individually:

$$\min_{h \in \mathcal{H}} \frac{1}{N} \sum_{i \in S} h_y(X_i)$$

$$\text{s.t} \quad \frac{1}{|G_{e,y}|} \sum_{i \in G_{e,y}} \mathbb{1}[h_y(X_i) = 0] \leq \gamma \quad \forall e \in \mathcal{E}_{train}. \quad (1)$$

Problem 1 is non-smooth and becomes computationally intractable for high dimensions and large datasets. To leverage standard gradient-based optimization methods, we introduce a relaxation of the optimization problem along with a parameterization of the predictor. Suppose the hypothesis class $\mathcal{H}$ is parameterized by $\theta \in \mathbb{R}^q$ (where $q$ may differ from the feature dimension $d$), such that $h_y(X) = 1$ if and only if $h_y^\theta(X) \geq 0$. This leads us to the following relaxed optimization problem:

$$\min_{\theta} \sum_{i \in S} \max\{0, 1 + h_y^\theta(X_i)\}$$

$$\text{s.t} \ \forall e \in E \ \frac{1}{|G_{e,y}|} \sum_{i \in G_{e,y}} \max\{0, 1 - h_y^\theta(X_i)\} \leq \gamma. \quad (2)$$

In appendix C we provide detailed derivations for this relaxation and show that its objective is a surrogate for the objective of problem (1), and that its constraints are also surrogate for those in the original formulation. We note that Wang & Qiao (2023b) derive a similar optimization problem for a setting of OOD class detection with two domains.

We solve problem 2 by introducing Lagrange multipliers $C \in \mathbb{R}^{m \times |\mathcal{Y}|}$ and define the Lagrangian:

$$L_y(\theta, C) = \sum_{i \in S} \max\{0, 1 + h_y^\theta(X_i)\} +$$

$$\sum_{e \in E_{train}} C_{e,y} \left( \frac{1}{|G_{e,y}|} \sum_{i \in G_{e,y}} \max\{0, 1 - h_y^\theta(X_i)\} - \gamma \right).$$

Gradient descent can then be performed on $\theta$, combined with a gradient ascent on C.

**To optimize** $\theta$, we focus on the relevant part of the Lagrangian:

$$L_y(\theta, C) = \sum_{i \in S} \max\{0, 1 + h_y^\theta(X_i)\} +$$
$$\sum_{e \in E_{train}} C_{e,y} \frac{1}{|G_{e,y}|} \sum_{i \in G_{e,y}} \max\{0, 1 - h_y^\theta(X_i)\}.$$

This can be simplified further by absorbing the $\frac{1}{|G_{e,y}|}$ factor into $C_{e,y}$, resulting in the following loss function for optimizing $\theta$:

$$L_y(\theta, C) = \sum_{i \in S} \max\{0, 1 + h_y^\theta(X_i)\} +$$
$$\sum_{e \in E_{train}} \mathbb{1}_{i \in G_{e,y}} C_{e,y} \max\{0, 1 - h_y^\theta(X_i)\}.$$

For wrong classification, the above loss will punish the predictor $h$. For correct classification, the second term will not punish it, but the first term will do (specifically for true positives). We found during experimentation that changing the first term to punish only wrong classifications helps stabilizing the learning process (Further details are given in Appendix E.6.2). Therefore we further adjust the loss to be

$$L_y(\theta, C) = \sum_{i \notin G_y} \max\{0, 1 + h_y^\theta(X_i)\} +$$
$$\sum_{e \in E_{train}} \mathbb{1}_{i \in G_{e,y}} C_{e,y} \max\{0, 1 - h_y^\theta(X_i)\}.$$

**To optimize** $C$, we suggest updating it once every few batches (the frequency is treated as a training hyperparameter) and at the end of each epoch. The updates are aimed to increase $C_{e,y}$ when the recall of $h_y^\theta$ in domain $e$ and label $y$ is lower than $1 - \gamma$, and decrease $C_{e,y}$ otherwise. The implementation details of these updates are outlined in Algorithm 1 (in Appendix D).

We call this approach SET Coverage Optimized with Empirical Robustness (SET-COVER) and outline it in Algorithm 1 (in Appendix D). In Section 5 we demonstrate the effectiveness of this algorithm through experiments on synthetic and real-world data.

### 4.2. Conformal Prediction Baselines

To evaluate the performance of SET-COVER, we present several set-prediction baselines that extend conformal prediction to DG and compare SET-COVER to them. Conformal prediction models have gained popularity in recent years for providing well-grounded prediction sets with strong marginal coverage guarantees (Gibbs & Candes, 2021; Romano et al., 2020; Tibshirani et al., 2019). The central idea behind conformal prediction for classification tasks is

to train a base model and use its predicted logits as "conformal scores" – a measure of how well input features $X$ relate to each of the labels in $\mathcal{Y}$. Then, a calibration process produces thresholds for these conformal scores, so that prediction sets that include labels for which the conformal scores fall within these thresholds guarantee a certain level of marginal coverage (Shafer & Vovk, 2008; Vovk et al., 2005). Conformal predictors have been widely used for providing set-valued predictions in single-domain settings, offering marginal coverage guarantees (Lei et al., 2013; Fontana et al., 2023).

#### 4.2.1. ROBUST CONFORMAL FOR DOMAIN GENERALIZATION

We suggest an adaptation of conformal prediction to provide robust performance across domains, calling it **Robust Conformal Prediction**:

- We first train a single classifier on the pooled data from all training domains and use its output logits as conformal scores for each label separately. We denote this as $f(x) = \langle f(x)_1, ..., f(x)_{|\mathcal{Y}|} \rangle$.

- For each training domain separately, we calibrate thresholds so that conformal scores within these thresholds provide the desired coverage in that specific domain. To achieve $1 - \gamma$ recall for each label in $\mathcal{Y}$ and for each training domain, we define the following set of thresholds:

$$t_{e,y} = \inf\{\alpha \; : \; \frac{1}{|G_{e,y}|} \sum_{i \in G_{e,y}} \mathbb{1}[f(X_i)_y > \alpha] \geq 1 - \gamma\}.$$

- At prediction time, a label is added to the prediction set if its conformal score exceeds the threshold in any of the training domains:
$$h_y(X_i) = \max_{e \in \mathcal{E}_{train}} \mathbb{1}[f(X_i)_y > t_{e,y}].$$

#### 4.2.2. POOLING CDFS

Dunn et al. (2018) proposed **Pooling-CDFs** for regression tasks, a DG approach based on conformal prediction. Unlike robust conformal methods, Pooling-CDFs aims to achieve average coverage across domains. In Pooling-CDFs a base model is trained on a pooled training data, gathered from several training domains. Then, the calibration is done on calibration data, gathered from different calibration domains, and thresholds are calculated to provide an average coverage guarantee across domains.

For our experiments, we adapt Pooling-CDFs to our min-recall objective by calculating thresholds for each label individually, aiming for 90% recall across domains. Additionally, we test a variant that performs both training and calibration on the full dataset available during training. We call this variant Pooling CDFs – Train Calibration (TrainC), and the original variant we call Pooling CDFs – Cross Vali-

dated Calibration (CVC).

# 5. Experiments

We conduct several experiments to evaluate the performance of five different approaches to the problem of DG. In line with the setup presented in previous sections, we aim to achieve a 90% recall for each label and each domain, while keeping the prediction set size as small as possible. We evaluate the following predictors:

**ERM**: We train a Neural Network (NN) to make single-valued predictions by minimizing a cross-entropy loss across the entire shuffled training dataset. This method represents the conventional method for training and deploying machine learning models.

**Pooling CDFs (TrainC)** and **Pooling CDFs (CVC)**: We implement the two variants of Pooling CDFs as described in section 4.2.2, building on the ERM classifier as the base model.

**Robust Conformal**: We implement a robust conformal predictor as described in Section 4.2.1, again building on the ERM classifier as the base model.

**SET-COVER**: We train a NN using the optimization process outlined in Section 4.1, with the aim of producing a model with a small prediction set size while maintaining the desired coverage.

Evaluating set-valued predictions presents challenges since models need to balance coverage with prediction set size. It also complicates model comparisons because it is unclear which metric should be prioritized. We choose to tackle this by presenting the following comparisons. First, we compare models on two key aspects: (1) the average size of prediction sets and (2) the minimum recall per domain. We visualize these results using cross-plots where the y-axis represents min-recall, and the x-axis represents average set size. Each cross shows the median and the 25th and 75th percentiles for both metrics across domains, as shown in Figures 1 and 2. Additionally, we present in Table 1 the percentage of domains achieving at least 90% min-recall, which is our predefined performance target. All our experiments in this section are repeated with five different random seeds, with results averaged across the seeds.

While the experiments in this section compare mainly set predictors (with ERM as the singleton-prediction baseline), in Appendix E.5 we also compare SET-COVER to few common singleton-prediction OOD baselines.

## 5.1. Synthetic Data

We first evaluate our proposed methods on synthetic data. We follow a data generation process proposed by Heinze-Deml & Meinshausen (2021), that incorporates both invariant features, i.e. features that do not depend on the domain, and domain-dependent features. However, unlike

Heinze-Deml & Meinshausen (2021)'s original setup, where invariant features alone were sufficient for optimal prediction, we adjusted the data generation process, ensuring that ignoring variant features would lead to a decline in performance.

The DGP is as follows:

$$Z_e \sim U[u_{\text{low}}, u_{\text{high}}] \quad ; \quad Y \sim Bernoulli(0.5)$$

$$X \sim Y(\mu + Z_e\nu) + N(0, \Sigma)$$

where $\mu, \nu \in \mathbb{R}^d$.

We experiment with $d = 10, 50$. For all methods we used a 2-layer multi-layer perceptron (MLP) as the base architecture. See details in Appendices E.1 and E.2.

The results in Figure 1 consistently show that both the robust conformal method and the SET-COVER method achieve the required coverage in test domains. As expected, the SET-COVER approach maintains a smaller average prediction set size.

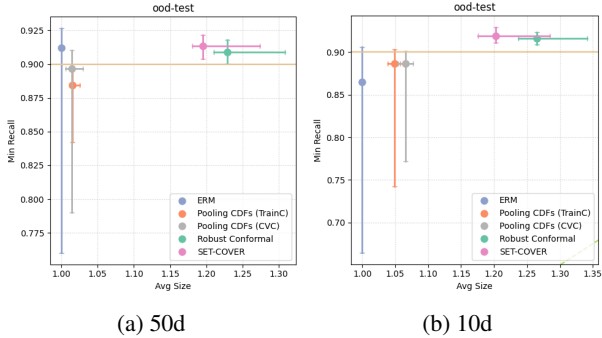

(a) 50d        (b) 10d

*Figure 1.* Min Recall distribution VS Mean Set Size distri- bution. **Blue** represents ERM predictor, **Orange** represents Pooling CDFs (TrainC), **Grey** represents Pooling CDFs (CVC), **Green** represents robust conformal predictor, and **Pink** represents SET-COVER. The horizontal solid line represents the 90% recall target value.

In Appendix E.3 we conduct experiments on a similar DGP where the covariance matrices vary randomly across domains. The results show that the performance of the models, including SET-COVER, remains stable even with domain-dependent covariance.

## 5.2. WILDS Data

We evaluate set-valued models on benchmarks from the WILDS (Koh et al., 2021) suite of benchmarks, which is designed to test models against real-world distribution shifts across various datasets and modalities. The benchmarks we use include:

**Camelyon** (Bandi et al., 2018): This dataset consists of pathological scans from 43 patients across 5 hospitals. Each

scan is divided into thousands of patches, each labeled as 1 if it contains a tumor and 0 otherwise. Patients are treated as domains, and the task of predicting whether a tumor appears in a patch (an image) is a binary classification.

**FMoW** (Christie et al., 2018): A satellite image dataset categorized by different land or building use types. Images come from different geographic areas and years, with each area-year pair defining a domain. We focus on the three most prevalent categories, constructing a multi-class classification task.

**iWildCam** (Beery et al., 2020): This dataset includes images of animals in the wild, taken from different locations, with the locations defining the domains. We again focus on the three most prevalent classes. The original images include a frame that reveals its domain, so we crop the images to discard these frames.

**Amazon** (Ni et al., 2019): A dataset of textual reviews, each is associated with a star rating from 1 to 5. The reviews are written by different reviewers, each has multiple reviews in the dataset. Each reviewer is considered a different domain, and the task is to predict the star rating of a review, i.e a multi-class classification task.

We discard data instances if they come from any (domain, label) pair with less than 100 samples. For Camelyon, FMoW, and iWildCam data sets, we use a pre-trained ResNet-18 network (He et al., 2016), fine-tuning it on the training set. For the Amazon dataset we use the embeddings of a pre-trained Bert (bert-base-uncased) model (Devlin et al., 2018) to calculate average embedding of the input text, and then pass it through a 2-layer MLP. All training sets are composed of randomly sampled instances from randomly sampled domains, and for all data sets the methods are evaluated on unseen test domains. Further experimental details are available at Appendix E.1. Training time comparisons are provided in Appendix E.6.3.

**Results** Figure 2 and Table 1 show consistently that both the robust conformal predictor and the SET-COVER predictor improve coverage in test domains, compared to the other methods. Furthermore, SET-COVER maintains a significantly smaller prediction set size compared to the robust conformal predictor. These results suggest that the SET-COVER method might be a promising step towards an efficient set-valued domain generalization.

*Table 1.* Summary of OOD Results on Wilds Datasets

| Model | Camelyon | | |
| --- | --- | --- | --- |
| | Median Min Recall ↑ | Median Avg Size ↓ | Recall ≥ 90% Pctg ↑ |
| ERM | $0.93 \pm 0.04$ | $1.0 \pm 0$ | $0.63 \pm 0.11$ |
| CDF Pooling-(TrainC) | $0.88 \pm 0.03$ | $1.07 \pm 0.15$ | $0.45 \pm 0.10$ |
| CDF Pooling-(CVC) | $0.81 \pm 0.14$ | $1.24 \pm 0.17$ | $0.33 \pm 0.19$ |
| Robust Conformal | $0.98 \pm 0.01$ | $1.79 \pm 0.16$ | $0.93 \pm 0.05$ |
| SET-COVER | $0.96 \pm 0.03$ | $1.05 \pm 0.03$ | $0.71 \pm 0.14$ |

| Model | FMoW | | |
| --- | --- | --- | --- |
| | Median Min Recall ↑ | Median Avg Size ↓ | Recall ≥ 90% Pctg ↑ |
| ERM | $0.76 \pm 0.06$ | $1.0 \pm 0.00$ | $0.08 \pm 0.06$ |
| CDF Pooling-(TrainC) | $0.81 \pm 0.01$ | $1.01 \pm 0.05$ | $0.07 \pm 0.04$ |
| CDF Pooling-(CVC) | $0.83 \pm 0.03$ | $1.10 \pm 0.02$ | $0.23 \pm 0.10$ |
| Robust Conformal | $0.89 \pm 0.01$ | $1.17 \pm 0.07$ | $0.43 \pm 0.15$ |
| SET-COVER | $0.91 \pm 0.02$ | $1.10 \pm 0.04$ | $0.72 \pm 0.22$ |

| Model | iWildCam | | |
| --- | --- | --- | --- |
| | Median Min Recall ↑ | Median Avg Size ↓ | Recall ≥ 90% Pctg ↑ |
| ERM | $0.99 \pm 0.00$ | $1.0 \pm 0.00$ | $0.71 \pm 0.03$ |
| CDF Pooling-(TrainC) | $0.92 \pm 0.03$ | $0.92 \pm 0.03$ | $0.53 \pm 0.06$ |
| CDF Pooling-(CVC) | $0.99 \pm 0.01$ | $1.14 \pm 0.12$ | $0.73 \pm 0.06$ |
| Robust Conformal | $0.99 \pm 0.01$ | $2.00 \pm 0.63$ | $0.86 \pm 0.07$ |
| SET-COVER | $0.99 \pm 0.00$ | $1.01 \pm 0.02$ | $0.82 \pm 0.09$ |

| Model | Amazon | | |
| --- | --- | --- | --- |
| | Median Min Recall ↑ | Median Avg Size ↓ | Recall ≥ 90% Pctg ↑ |
| ERM | $1.0 \pm 0.00$ | $1.0 \pm 0.00$ | $0.68 \pm 0.06$ |
| CDF Pooling-(TrainC) | $0.94 \pm 0.01$ | $4.31 \pm 0.17$ | $0.63 \pm 0.02$ |
| CDF Pooling-(CVC) | $0.99 \pm 0.00$ | $2.48 \pm 0.53$ | $0.70 \pm 0.06$ |
| Robust Conformal | $1.0 \pm 0.00$ | $4.68 \pm 0.28$ | $0.99 \pm 0.00$ |
| SET-COVER | $1.0 \pm 0.00$ | $2.43 \pm 0.09$ | $0.96 \pm 0.02$ |

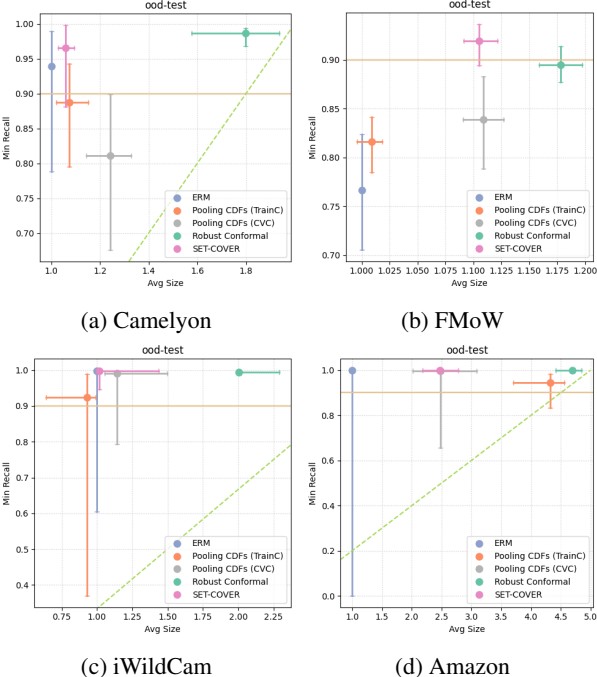

(a) Camelyon

(b) FMoW

(c) iWildCam

(d) Amazon

*Figure 2.* Each figure represents Min-Recall over Avg Set Size cross. y-axis represents min-recall, and x-axis represents average set size. Each cross shows the median and the 25th and 75th percentiles for both metrics across domain. **Blue** represents ERM predictor, **Orange** represents Pooling CDFs (TrainC), **Grey** represents Pooling CDFs (CVC), **Green** represents robust conformal predictor, and **Pink** represents SET-COVER. The horizontal solid line represents the 90% recall target value, and dashed yellow diagonal line represents performance of a random predictor.

## Reproducibility Code

We release all code and evaluation scripts at https://github.com/ront65/set-valued-ood to facilitate reproducibility.

## Acknowledgments

We would like to thank the anonymous reviewers of the paper for their helpful comments and suggestions. RT and US were supported by ISF grant 2456/23. RT and DN were supported by ISF grant 827/21.

## Impact Statement

This work proposes a novel approach to Out of Distribution (OOD) generalization by introducing set-valued predictors to tackle the challenges of domain shifts. The proposed method emphasizes robust, worst-case performance guarantees across unseen domains, which is crucial for high-stakes applications like healthcare. By advancing the understanding of domain generalization and providing practical tools for set-valued prediction, this research contributes to improving the reliability of machine learning systems in critical real-world settings.

As this paper presents work whose goal is to advance the field of Machine Learning, there are many potential societal consequences of this work, none which we feel must be specifically highlighted here.

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

# A. Proofs For Theoretical Claims

## A.1. Proof of Theorem 3.4

The definitions we provide for uniform convergence and shattering in the multi-domain setting differ slightly from those in classical machine learning. First, in classical ML, the connection between VC-dimension and uniform convergence is typically defined with respect to the 0-1 loss, whereas in our case, it is defined in terms of a performance indicator. Second, while classical ML defines uniform convergence with respect to distributions over $\mathcal{X} \times \{0, 1\}$, in our setting, it is defined over distributions on $\mathcal{E}$ alone. These distinctions require careful consideration when linking VC-dimension to uniform convergence in the multi-domain framework, as we do in Theorem 3.4. We will now address this and provide a proof for the theorem.

**Theorem 3.4.** *A hypothesis class $\mathcal{H}$ has the uniform convergence property with respect to a loss function $\mathcal{L}$ and threshold $\gamma$ if and only if $VCdim_{\mathcal{L},\gamma}(\mathcal{H}) < \infty$. Moreover, if $d = VCdim_{\mathcal{L},\gamma}(\mathcal{H}) < \infty$, the sample size from the uniform-convergence definition is $m(\delta, \epsilon) = \Theta(\frac{d + log(1/\delta)}{\epsilon^2})$.*

*Proof.* **First Direction:** We start by proving that if $\mathcal{H}$ has the uniform convergence property, then $VCdim_{\mathcal{L},\gamma}(\mathcal{H}) < \infty$. Assume $\mathcal{H}$ possesses the uniform convergence property. This means that for any $0 < \epsilon, \delta < 0.5$ there exists $m(\epsilon, \delta)$ such that for any distribution $D$ over $\mathcal{E}$ and for any training sample $S \subset \mathcal{E}$ drawn according to $D$ with $|S| \geq m(\epsilon, \delta)$ the probability (over the draw of $S$) that

$$\exists h \in \mathcal{H} \; s.t \left| \mathop{\mathbb{E}}_{e \sim D}[\mathbb{1}_{\mathcal{L},\gamma}(h, e)] - \frac{1}{|S|} \sum_{e \in S} \mathbb{1}_{\mathcal{L},\gamma}(h, e) \right| > \epsilon$$

is less than $\delta$.

Now, suppose by contradiction that $VCdim_{\mathcal{L},\gamma}(\mathcal{H}) = \infty$. This means that there exist a set $C \subset \mathcal{E}$ with $|C| = 2m(\epsilon, \delta)$ that can be shattered by $\mathcal{H}$. Let us define a distribution $D$ over $\mathcal{E}$ such that:

$$D(e) = \begin{cases} \frac{1}{|C|} & \text{if } e \in C \\ 0 & otherwise \end{cases}$$

Since $\mathcal{H}$ shatters $C$, for any subset $S \subset C$ with $|S| = m(\epsilon, \delta) = \frac{|C|}{2}$ there exist $h_s \in \mathcal{H}$ such that:

$$\mathbb{1}_{\mathcal{L},\gamma}(h_s, e) = \begin{cases} 0 & \text{if } e \in S \\ 1 & \text{else} \end{cases}$$

For this hypothesis $h_s$, we have

$$\frac{1}{|S|} \sum_{e \in S} \mathbb{1}_{\mathcal{L},\gamma}(h_s, e) = 0 \quad \text{and} \quad \mathop{\mathbb{E}}_{e \sim D}[\mathbb{1}_{\mathcal{L},\gamma}(h_s, e)] = 0.5$$

Thus, for a specific choice of $S$ we have shown that:

$$\exists h_s \in \mathcal{H} \; . \; | \mathop{\mathbb{E}}_{e \sim D}[\mathbb{1}_{\mathcal{L},\gamma}(h, e)] - \frac{1}{|S|} \sum_{e \in S} \mathbb{1}_{\mathcal{L},\gamma}(h, e)| = 0.5 > \epsilon.$$

Since this holds for any sample $S$, the probability of the above condition occuring is $1 > \delta$, which contradicts the uniform convergence property. Therefore, our assumption that $VCdim_{\mathcal{L},\gamma}(\mathcal{H}) = \infty$ must be false.

**Second direction:** Next, we will prove that if $VCdim_{\mathcal{L},\gamma}(\mathcal{H}) < \infty$, then $\mathcal{H}$ has the uniform convergence property.

Let $0 < \epsilon, \delta < 1$. Given a hypothesis set $\mathcal{H}$, loss function $\mathcal{L}$ and $0 \leq \gamma \leq 1$ we define a new hypothesis set $\mathcal{H}^*$ as follows:

$$\mathcal{H}^* = \big\{ \mathbb{1}_{\mathcal{L},\gamma}(h, \cdot) \; : \; h \in \mathcal{H} \big\}.$$

Each hypothesis $\mathbb{1}_{\mathcal{L},\gamma}(h, \cdot)$ in $\mathcal{H}^*$ maps elements of $\mathcal{E}$ to $\{0, 1\}$.

It is straight forward from the definitions that if $VCdim_{\mathcal{L},\gamma}(\mathcal{H}) < \infty$ according to definition 3.2 of VCdim, than $VCdim(\mathcal{H}^*) < \infty$ according to the classical definition of VCdim, and that both VCdim values are equal (we denote

them as $d$). By classical results on VCdim, this implies that $\mathcal{H}^*$ has the uniform convergence property, with sample size $m(\delta, \epsilon) = \Theta(\frac{d + log(1/\delta)}{\epsilon^2})$.

We would like to argue that if $\mathcal{H}^*$ has the uniform convergence property (according to the classical definition), so does $\mathcal{H}$ according to our definition 3.3. However, there are two subtle differences in the definitions that require special care.

- First, the uniform convergence of $\mathcal{H}^*$ is with regard to destributions $D^*$ over $\mathcal{E} \times \{0, 1\}$. Therefore we will translate our distribution $D$, which is over $\mathcal{E}$, to a distribution $D^*$ as required, and explain the connection between them.

- Second, the uniform convergence of $\mathcal{H}^*$ is with regard to $\mathcal{L}_{0-1}$ loss. Therefore we will need to connect the $\mathcal{L}_{0-1}$ loss of $\mathcal{H}^*$ to our expression of interest, $\mathbb{1}_{\mathcal{L}, \gamma}(h, e)$.

We will now tackle these two points to prove uniform convergence of $\mathcal{H}$.

First, let $D$ be a distribution over $\mathcal{E}$, and define $D^*$ to be a distribution over the space $\mathcal{E} \times I$ where $I = \{0, 1\}$ such that:

$$D^*(e, i) = \begin{cases} D(e) & \text{if } i = 0 \\ 0 & \text{else} \end{cases}$$

In words, $D^*$ has the same density over $\mathcal{E}$ as $D$ when $i = 0$, and zero density otherwise.

By the uniform convergence property of $\mathcal{H}^*$, for a sample $S \subset \mathcal{E} \times I$ with $|S| \geq m(\epsilon, \delta) = \Theta(\frac{d + log(1/\delta)}{\epsilon^2})$ drawn according to $D^*$, it holds with probability at least $1 - \delta$ over $S$ that :

$$\forall\, \mathbb{1}_{\mathcal{L}, \gamma}(h, \cdot) \in \mathcal{H}^* \, \Big|\, \mathbb{E}_{e, i \sim D^*} \big[\mathcal{L}_{0-1}\big(\mathbb{1}_{\mathcal{L}, \gamma}(h, e), i\big)\big] - \frac{1}{|S|} \sum_{e, i \in S} \mathcal{L}_{0-1}\big(\mathbb{1}_{\mathcal{L}, \gamma}(h, e), i\big)\Big| \leq \epsilon$$

Since $D^*$ is constructed such that $i = 0$ always, we have :

$$\mathcal{L}_{0-1}[\mathbb{1}_{\mathcal{L}, \gamma}(h, e), i] = \mathbb{1}_{\mathcal{L}, \gamma}(h, e)$$

thus, we can rewrite the above as:

$$\forall\, h \in \mathcal{H} \,.\, \big|\, \mathbb{E}_{e \sim D}[\mathbb{1}_{\mathcal{L}, \gamma}(h, e)] - \frac{1}{|S|} \sum_{e \in S} \mathbb{1}_{\mathcal{L}, \gamma}(h, e)\big| \leq \epsilon$$

Overall we have shown that whenever $|S| \geq m(\epsilon, \delta) = \Theta(\frac{d + log(1/\delta)}{\epsilon^2})$, for any distribution $D$ over $\mathcal{E}$ the above condition holds with probability at least $1 - \delta$, which precisely matches the definition of uniform convergence as outlined in Definition 3.3. $\square$

### A.2. Proof of Theorem 3.5
**Theorem 3.5.** *For $d = 1$, linear $\mathcal{H}$ cannot shatter more than $2|\mathcal{Y}|$ domains with respect to $\mathcal{L}_{recall}$ and any $0 < \gamma < 1$.*

*Proof.* Let $X \in R$ and $y \in \mathcal{Y}$. Consider any set of n domains: $\hat{\mathcal{E}} = \{e_1, e_2, \dots e_n\}$.

A linear classifier $h_y$ is of one of the forms:

$$h_y(x) = \begin{cases} 0 & \text{if } x \geq a \\ 1 & \text{otherwise} \end{cases} \qquad h_y(x) = \begin{cases} 0 & \text{if } x < a \\ 1 & \text{otherwise} \end{cases}.$$

We start by showing that for each of these options, there is a domain in $\hat{\mathcal{E}}$ such that if $h_y$ archives a "good" coverage in that domain (i.e $p_e[h_y(X) = 1 | Y = y] \geq 1 - \gamma$), it must also achieve such coverage in all $\hat{\mathcal{E}}$. From this, we willl conclude that there exists a subset of domains in $\hat{\mathcal{E}}$ such that achieving a "good" recall-loss within this subset will necessarily result in a "good" recall-loss across all of $\hat{\mathcal{E}}$.

For each domain we can consider the conditional CDF $F_{e_i}|_{Y=y}(x) = P^{e_i}[X < x | Y = y]$. Note that $F_{e_i}^{-1}|_{Y=y}(q)$ is the value $x \in R$ such that $P^{e_i}[X < x | Y = y] = q$.

Assume $h_y$ is of the form

$$h_y(x) = \begin{cases} 0 & \text{if } x \geq a \\ 1 & \text{otherwise} \end{cases}$$

Let us define:

$$i_{max,y} = \arg\max_{i \in [n]} F_{e_i}^{-1}|_{Y=y}(1-\gamma)$$

Now assume that:

$$p_{e_{i_{max,y}}}[h_y(X) = 1|Y = y] \geq 1 - \gamma$$

Then for each $i \in [n]$ is holds that

$$a \geq F_{e_{i_{max,y}}}^{-1}|_{Y=y}(1-\gamma) \geq F_{e_i}^{-1}|_{Y=y}(1-\gamma) \implies p_{e_i}[h_y(X) = 1|Y = y] \geq 1 - \gamma$$

In a similar way, if $h_y$ is of the form

$$h_y(x) = \begin{cases} 0 & \text{if } x < a \\ 1 & \text{otherwise} \end{cases}$$

Let us define:

$$i_{min,y} = \arg\min_{i \in [n]} F_{e_i}^{-1}|_{Y=y}(\gamma)$$

and assume that:

$$p_{e_{i_{min,y}}}[h_y(X) = 1|Y = y] \geq 1 - \gamma$$

Then for each $i \in [n]$ is holds that

$$a \leq F_{e_{i_{min,y}}}^{-1}|_{Y=y}(\gamma) \leq F_{e_i}^{-1}|_{Y=y}(\gamma) \implies p_{e_i}[h_y(X) = 1|Y = y] \geq 1 - \gamma$$

We see that whenever the recall condition, given $Y = y$, holds for both $i_{min,y}$ and $i_{max,y}$, then it holds for all $i \in [n]$. Now, if $n > 2|\mathcal{Y}|$, there must be some $i \in [n]$ such that $i \neq i_{min,y}$ and $i \neq i_{max,y}$ for any $y \in \mathcal{Y}$.
From the above results, it holds that it is not possible to achieve any assignment such that

$$\forall_{y \in \mathcal{Y}} \; \mathbb{1}_{\mathcal{L}_{recall,\gamma}}(h, e_{i_{min,y}}) = 1$$
$$\forall_{y \in \mathcal{Y}} \; \mathbb{1}_{\mathcal{L}_{recall,\gamma}}(h, e_{i_{max,y}}) = 1$$
$$\mathbb{1}_{\mathcal{L}_{recall,\gamma}}(h, e_i) = 0$$

Therefore, linear classifiers cannot shatter any set of $n > 2|\mathcal{Y}|$ domains in the case of $\mathcal{X} \subseteq R$.

$\square$

### A.3. Prof of Theorem 3.6
**Theorem 3.6.** *For $d > 1$, and any size $n$, there exists a set of $n$ domains that can be shattered by linear hypotheses $\mathcal{H}$ with respect to $\mathcal{L}_{recall}$ and any $0 < \gamma < 1$.*

*Proof.* To prove this, it is enough to consider the case of $d = 2$, as for $d > 2$ we can always consider domains that represent degenerate distributions on a $2d$ subspace with linear classifiers that practically operate in that subspace.

In $R^2$ we can consider domains such that their feature distributions (i.e the marginal distributions over $\mathcal{X}$) have a support on the unit circle, and linear classifiers that pass through the origin. It is not hard to see that in that case the problem reduces to distributions in the interval $[0, 2\pi] \subset R$ with classifiers that are rectangles of length $\pi$. In Lemma A.1 we prove that for any $n \in \mathbf{N}$ there are $n$ domains that can be shattered by rectagles. That proof also holds when we limit the distributions to be on a closed interval $I \subset R$, with classifiers that are rectangles of a fixed length (no matter what the length is, as long as it is smaller than the length of I). Therefore, the same proof holds in our reduced case. $\square$

**Lemma A.1.** *For any $d$ and any size $n$, there is a set of $n$ domains that can be shattered by rectangular $\mathcal{H}$ with respect to $\mathcal{L}_{recall}$ and any $0 < \gamma < 1$.*

*Proof.* For a given $n \in \mathcal{N}$ we need to present a set of $n$ domain $C = \{e_1, ..., e_n\}$ such that:

$$\forall_{I \subseteq [n]} \exists_{h \in \mathcal{H}_{rect}} \ s.t \ \forall_{y \in \mathcal{Y}} P_{e_i}[y \in h(X)|Y = y] \geq 1 - \gamma \iff i \in I$$

We consider few simplifications for the proof:

- First, it is enough to show the above for $d = 1$. This is because for $d > 1$ we can always define the domains in $C$ to be degenerate distributions on a $1d$ subspace, and then follow the same construction as will be presented here for $d = 1$.

- We can define the domains in $C$ to be such that $P_e[X|Y = y]$ will not depend on the value of $y$. Then, because $h$ can be break down to $h_y$, it is enough to show that for some $y \in \mathcal{Y}$

$$\forall_{I \subseteq [n]} \exists_{h_y \in \mathcal{H}_{y-rect}} \ s.t \ P_{e_i}[y \in h(X)|Y = y] \geq 1 - \gamma \iff i \in I$$

  Where $\mathcal{H}_{y-rect}$ is the set of rectangular hypothesis for $h_y$. This is true because we can use the same function $h_y$ for all values of $y$ to get the original goal.

We will now define distributions for $X|Y = y$ for the domains in $C$ and show that our simplified goal holds.

Consider some order on all the subsets of $[n]$ such that the first subset is the empty one:

$$I_0 = \emptyset, I_1, ..., I_{2^n} \in \mathcal{P}([n])$$

We need to construct domains for $C$ and show that for each $0 \leq j \leq 2^n - 1$ there is a rectangular classifier $h_y^j$ such that

$$P_{e_i}[h_y^j(X) = 1|Y = y] \geq 1 - \gamma \iff i \in I_j$$

The idea of our construction is as follows. We will define the density of $X$ on the interval $(0, 3)$. We split the intervals $(0, 1)$ and $(2, 3)$ to $2^n$ sub-intervals, and recognize these sub-intervals with the subgroups $I$ of $[n]$. Then, we will look at a sliding window, such that each progress of the window adds one sub-interval from $(2, 3)$ to the cover, and discards one interval from $(0, 1)$. We will spread the density between the sub-intervals such that the sub-interval that is being discarded causes the associated domains to loss cover, while the sub-interval that is being included causes the associated domains to gain required coverage. This way our sliding window represents rectangular classifies that can achieve any assignment for the domains.

The formal construction is as follows:

$$P_{e_i}[X|Y = y] = \begin{cases} \frac{2^n}{2^{n-1}+1}0.01(1 - \gamma) & \text{if } \exists_{0 \leq j \leq 2^n-1} \frac{j}{2^n} \leq X \leq \frac{j+1}{2^n} \ \& \ i \in I_j \\ 0.99(1 - \gamma) & \text{if } 1 \leq X \leq 2 \\ \frac{2^n}{2^{n-1}+1}0.01(1 - \gamma) & \text{if } \exists_{1 \leq j \leq 2^n-1} (2 + \frac{j-1}{2^n}) \leq X \leq (2 + \frac{j}{2^n}) \ \& \ i \in I_j \\ a & \text{if } -2 \leq X \leq -1 \\ 0 & \text{otherwise} \end{cases}$$

Where we assume that $1 - \gamma$ is small enough such that the total integral for $X > 0$ is less than 1, and $a$ completes the density to totally integral to 1. If $1 - \gamma$ is not small enough we just need to modify the factors 0.99 and 0.01 of the density, so WLOG we assume $1 - \gamma$ is small enough.

We will now prove by induction that for $0 \leq j \leq 2^n - 1$ we have:

$$i \in I_j \implies P_{e_i}\left[\frac{j}{2^n} \leq X \leq 2 + \frac{j}{2^n}\Big|Y = y\right] = 1 - \gamma$$

$$i \notin I_j \implies P_{e_i}\left[\frac{j}{2^n} \leq X \leq 2 + \frac{j}{2^n}\Big|Y = y\right] = \frac{2^{n-1}}{2^{n-1}+1}0.01(1 - \gamma) + 0.99(1 - \gamma) < 1 - \gamma$$

Therefore, for the rectangular binary classifier $h_y^j(X) = 1 \iff \frac{j}{2^n} \leq X \leq 2 + \frac{j}{2^n}$ we have

$$P_{e_i}[h_y^j(X) = 1|Y = y] \geq 1 - \gamma \iff i \in I_j$$

This will conclude the proof, as it means that rectangular classifiers can shatter the set $C$ as required.

**Induction base case, j=0**. Each domain takes part in $2^{n-1}$ of the subgroups $I$, therefore from the domains construction the followng holds for each domain:

$$P_{e_i}[0 \leq X \leq 1|Y = y] = 2^{n-1}\frac{1}{2^n}\frac{2^n}{2^{n-1}+1}0.01(1-\gamma) = \frac{2^{n-1}}{2^{n-1}+1}0.01(1-\gamma)$$

Therefore for each domain $e_i$,

$$P_{e_i}\left[\frac{j}{2^n} \leq X \leq 2+\frac{j}{2^n}\middle|Y = y\right] = P_{e_i}[0 \leq X \leq 2|Y = y] = \frac{2^{n-1}}{2^{n-1}+1}0.01(1-\gamma) + 0.99(1-\gamma) < 1 - \gamma$$

Together with the fact that $I_0 = \emptyset$ this proves the base case.

**Induction step, j $\longrightarrow$ j+1**. From the induction assumption we know that

$$i \in I_j \Longrightarrow P_{e_i}\left[\frac{j}{2^n} \leq X \leq 2+\frac{j}{2^n}\middle|Y = y\right] = 1 - \gamma$$

$$i \notin I_j \Longrightarrow P_{e_i}\left[\frac{j}{2^n} \leq X \leq 2+\frac{j}{2^n}\middle|Y = y\right] = \frac{2^{n-1}}{2^{n-1}+1}0.01(1-\gamma) + 0.99(1-\gamma)$$

From the domains construction we have:

$$i \in I_j \Longrightarrow P_{e_i}\left[\frac{j}{2^n} \leq X \leq \frac{j+1}{2^n}\middle|Y = y\right] = \frac{1}{2^n}\frac{2^n}{2^{n-1}+1}0.01(1-\gamma)$$

$$i \notin I_j \Longrightarrow P_{e_i}\left[\frac{j}{2^n} \leq X \leq \frac{j+1}{2^n}\middle|Y = y\right] = 0$$

Putting these together will give us:

$$i \in I_j \Longrightarrow P_{e_i}\left[\frac{j+1}{2^n} \leq X \leq 2+\frac{j}{2^n}\middle|Y = y\right] = 1 - \gamma - \frac{1}{2^n}\frac{2^n}{2^{n-1}+1}0.01(1-\gamma) =$$

$$= 0.99(1-\gamma) + 0.01(1-\gamma) - \frac{1}{2^{n-1}+1}0.01(1-\gamma) = \frac{2^{n-1}}{2^{n-1}+1}0.01(1-\gamma) + 0.99(1-\gamma)$$

$$i \notin I_j \Longrightarrow P_{e_i}\left[\frac{j+1}{2^n} \leq X \leq 2+\frac{j}{2^n}\middle|Y = y\right] = \frac{2^{n-1}}{2^{n-1}+1}0.01(1-\gamma) + 0.99(1-\gamma)$$

i.e, we have for each domain

$$P_{e_i}\left[\frac{j+1}{2^n} \leq X \leq 2+\frac{j}{2^n}\middle|Y = y\right] = \frac{2^{n-1}}{2^{n-1}+1}0.01(1-\gamma) + 0.99(1-\gamma)$$

Now, from the construction of the domains we have

$$i \in I_{j+1} \Longrightarrow P_{e_i}\left[2+\frac{j}{2^n} \leq X \leq 2+\frac{j+1}{2^n}\middle|Y = y\right] = \frac{1}{2^n}\frac{2^n}{2^{n-1}+1}0.01(1-\gamma)$$

$$i \notin I_{j+1} \Longrightarrow P_{e_i}\left[2+\frac{j}{2^n} \leq X \leq 2+\frac{j+1}{2^n}\middle|Y = y\right] = 0$$

Therefore, we have

$$i \in I_{j+1} \Longrightarrow P_{e_i}\left[\frac{j+1}{2^n} \leq X \leq 2+\frac{j+1}{2^n}\middle|Y = y\right] =$$

$$= \frac{2^{n-1}}{2^{n-1}+1}0.01(1-\gamma) + 0.99(1-\gamma) + \frac{1}{2^n}\frac{2^n}{2^{n-1}+1}0.01(1-\gamma) =$$

$$= \frac{2^{n-1}+1}{2^{n-1}+1}0.01(1-\gamma) + 0.99(1-\gamma) = 1 - \gamma$$

$$i \notin I_{j+1} \Longrightarrow P_{e_i}\left[\frac{j+1}{2^n} \leq X \leq 2+\frac{j+1}{2^n}\middle|Y = y\right] = \frac{2^{n-1}}{2^{n-1}+1}0.01(1-\gamma) + 0.99(1-\gamma) + 0$$

That completes our induction. $\qquad\square$

### A.4. Proof of Theorem 3.7

**Theorem 3.7.** *If domains are restricted to being conditionally Gaussian, and there exists $\forall_{y \in \mathcal{Y}} \; \Sigma_y \in \mathbb{R}^{d \times d}$ such that for each $e, y \; \Sigma_{e,y} = \sigma_{e,y} \Sigma_y$ for some $\sigma_{e,y} \in \mathbb{R}$, then for each $\epsilon, \delta > 0$ there exist $m = \Theta\left(\frac{|\mathcal{Y}|^2 (d + log(|\mathcal{Y}|/\delta))}{\epsilon^2}\right)$ such that if $|\mathcal{E}_{train}| \geq m$ than with probability higher than $1 - \delta$ over the training domains sample $\mathcal{E}_{train}$, and regardless of the distribution $D$ over $\mathcal{E}$, any linear hypothesis $h$ such that*

$$\forall_{e \in \mathcal{E}_{train}} \; \mathbb{1}_{\mathcal{L}_{recall}, \gamma}(h, e) = 0$$

*will also have*

$$\mathbb{E}_{e \sim D}[\mathbb{1}_{\mathcal{L}_{recall}, \gamma}(h, e)] \leq \epsilon.$$

*Proof.* We begin by reminding that for a set-valued $h$ to be a linear hypothesis means it can be decomposed into $|\mathcal{Y}|$ linear binary classifiers $h_y$. We will break the proof into two stages. First, we define for each $y \in \mathcal{Y}$ a loss $\mathcal{L}_y(h_y, e) = P_e[h_y(X) = 0 | Y = y]$ and show that linear binary classifiers cannot shatter more than $d + 1$ domains with regard to $\mathcal{L}_y, \gamma$. Then, we will use this result to show that linear binary classifiers have the uniform convergence property with regard to $\mathcal{L}_y, \gamma$, and with that in hand we will conclude the theorem.

**Lemma A.2.** *In the terms of theorem A.2, linear binary classifiers cannot shatter more than $d + 1$ domains with respect to $\mathcal{L}_y$ and any $0 < \gamma < 1$*

We start by proving the Leamma.

As this Lemma focuses on a specific value of $y$, we will omit $y$ from some notations where it is clear from the context.

We assume $h_y$ is a linear binary classifiers, i.e there exists $\theta \in R^d$ such that

$$h_y(X) = 1 \iff \theta^T X \geq 0$$

Let $C = \{e_1, e_2, ..., e_n\}$ be a set of $n = d + 2$ domains that are conditionally normal, i.e

$$\forall 1 \leq i \leq n \quad [X | Y = y] \sim^{P_{e_i}} N(\mu_i, \sigma_i \cdot \Sigma).$$

We will mark

$$\mathbb{1}_i = 1 \text{ if } P_{e_i}[h_y(X) = 1 | Y = y] \geq 1 - \gamma \quad \text{else } 0$$

and show that there must be an assignment of $\{\mathbb{1}_i\}_{i=1}^n \in \{0, 1\}^n$ that cannot be achieved.

Given $\theta \in R^d$ let us consider the variable $Z = \theta^T X$. For each domain, this is a conditionally normal 1d variable:

$$[Z | Y = y] \sim^{e_i} N(\theta^T \mu_i, \; \sigma_i \theta^T \Sigma \theta).$$

Therefore,

$$P^{e_i}[Z > 0 | Y = y] = P^{e_i}\left[\frac{Z - \theta^T \mu_i}{\sqrt{\sigma_i \theta^T \Sigma \theta}} \geq -\frac{\theta^T \mu_i}{\sqrt{\sigma_i \theta^T \Sigma \theta}} \Big| Y = y\right] =$$

$$= \phi\left(\frac{\theta^T \mu_i}{\sqrt{\sigma_i \theta^T \Sigma \theta}}\right)$$

where $\phi(\cdot)$ is the CDF of a standaridized normal variable. Letting $c$ be such that $\phi(c) = 1 - \gamma$, we have

$$\mathbb{1}_i = 1 \iff P^{e_i}[Z > 0 | Y = y] \geq 1 - \gamma \iff \frac{\theta^T \mu_i}{\sqrt{\sigma_i \theta^T \Sigma \theta}} \geq c \iff \theta^T \mu_i - c\sqrt{\sigma_i \theta^T \Sigma \theta} \geq 0.$$

Defining $\hat{\theta} = (\theta, c\sqrt{\theta^T \Sigma \theta}) \in R^{d+1}$, $\nu_i = (\mu_i, \sqrt{\sigma_i}) \in R^{d+1}$ we get

$$\mathbb{1}_i = 1 \iff \hat{\theta}^T \nu_i \geq 0$$

We note that the fact that $\hat{\theta}$ does not depend on i at all, and that $\nu_i$ does not depend on $\theta$ is important, as will be seen next. Now, as we have $d + 2$ domains in $C$, we have $d + 2$ corresponding vectors $\nu_i$ in $R^{d+1}$, so there must be one vector that is a non-trivial linear combination of the others. WLOG assume that vector is $\nu_1$:

$$\nu_1 = \sum_{i>1} \alpha_i \nu_i$$

Now, let us consider the following assignment for $\mathbb{1}_i$:

$$\mathbb{1}_i = \begin{cases} 1 & \text{if } \alpha_i \geq 0 \\ 0 & \text{if } \alpha_i < 0 \\ 0 & i = 1 \end{cases}$$

Assuming there is a $\hat{\theta}$ that achieves this assignment would mean that:

$$i > 1 \implies \alpha_i \hat{\theta}^T \nu_i \geq 0$$

$$\hat{\theta}^T \nu_1 = \sum_{i>1} \alpha_i \hat{\theta}^T \nu_i \geq 0 \implies \mathbb{1}_1 = 1$$

which contradicts the requirement of $\mathbb{1}_1 = 0$. As $\hat{\theta}$ depends solely on $\theta$ (and on $c, \Sigma$, which are constants), and does not depend on $i$, this means that $\theta$ cannot satisfy the above assignment. As the suggested assignment is not dependent on $\theta$ (it depends only on the set of $\{\nu_i\}$, which are not dependent on $\theta$), it means that we have found an assignment that no $\theta$ can satisfy.

We have shown that no $\theta$ can satisfy the above assignment, which means linear classifiers are not able to shatter the set $C$, as required.

This proves the Lemma. Now, consider $\epsilon, \delta$ from the theorem statement. An immediate result of Lemma A.2 and theorem 3.4 is that, focusing on a single $y \in \mathcal{Y}$, the hypothesis set $\mathcal{H}_y$ of linear binary classifiers have the uniform convergence property with regard to $\mathcal{L}_y, \gamma$, i.e there exists $m_y(\frac{\epsilon}{|\mathcal{Y}|}, \frac{\delta}{|\mathcal{Y}|}) = \Theta(\frac{|\mathcal{Y}|^2(d+log(|\mathcal{Y}|/\delta))}{\epsilon^2})$ such that with probability higher than $1 - \frac{\delta}{|\mathcal{Y}|}$

$$|S| > m_y(\frac{\epsilon}{|\mathcal{Y}|}, \frac{\delta}{|\mathcal{Y}|}) \longrightarrow \forall h_y \in \mathcal{H}_y \ \mathop{\mathbb{E}}_{e \sim D}[\mathbb{1}_{\mathcal{L}_y,\gamma}(h, e)] < \frac{1}{|S|}\sum_{e \in S}[\mathbb{1}_{\mathcal{L}_y,\gamma}(h, e)] + \frac{\epsilon}{|\mathcal{Y}|}.$$

If we take $m = \max_{y \in \mathcal{Y}} m_y(\frac{\epsilon}{|\mathcal{Y}|}, \frac{\delta}{|\mathcal{Y}|}) = \Theta(\frac{|\mathcal{Y}|^2(d+log(|\mathcal{Y}|/\delta))}{\epsilon^2})$ than the above holds simultaneously for all $y \in \mathcal{Y}$ with probability higher than $1 - \delta$.

Now, we note that $\mathcal{L}_{recall}(h, e) = \max_{y \in \mathcal{Y}} \mathcal{L}_y(h, e)$, therefore if $|S| > m$ we have with proaility at least $1 - \delta$

$$\forall_{h \in \mathcal{H}} \ \mathop{\mathbb{E}}_{e \sim D}[\mathbb{1}_{\mathcal{L}_{recall},\gamma}(h, e)] = \mathop{P}_{e \sim D}[\mathcal{L}_{recall}(h, e) > \gamma] \leq \sum_{y \in \mathcal{Y}} \mathop{P}_{e \sim D}[\mathcal{L}_y(h, e) > \gamma] \leq$$

$$\leq \sum_{y \in \mathcal{Y}} (\frac{1}{|S|}\sum_{e \in S}[\mathbb{1}_{\mathcal{L}_y,\gamma}(h, e)] + \frac{\epsilon}{|\mathcal{Y}|}) \leq \frac{1}{|S|}\sum_{e \in S}\sum_{y \in \mathcal{Y}}[\mathbb{1}_{\mathcal{L}_y,\gamma}(h, e)] + \epsilon$$

And finally, as

$$\mathbb{1}_{\mathcal{L}_{recall},\gamma}(h, e) = 0 \longrightarrow \forall_{y \in \mathcal{Y}} \mathbb{1}_{\mathcal{L}_y,\gamma}(h, e) = 0$$

for any $h$ such that

$$\forall_{e \in \mathcal{E}_{train}} \mathbb{1}_{\mathcal{L}_{recall},\gamma}(h, e) = 0$$

it also holds that

$$\mathop{\mathbb{E}}_{e \sim D}[\mathbb{1}_{\mathcal{L}_{recall},\gamma}(h, e)] \leq \frac{1}{|S|}\sum_{e \in S}\sum_{y \in \mathcal{Y}}[\mathbb{1}_{\mathcal{L}_y,\gamma}(h, e)] + \epsilon \leq \frac{1}{|S|}\sum_{e \in S}\sum_{y \in \mathcal{Y}} 0 + \epsilon \leq \epsilon$$

$\square$

# B. Sample Size Analysis

In this section, we examine the sample size requirements for achieving generalization from training domains with finite samples to new, unseen domains. We start with some notations:

- We recall the definition of performance indicator: $\mathbb{1}_{\mathcal{L},\gamma}(h, e) = \mathbb{1}[\mathcal{L}(h, e) \geq \gamma]$

- We denote the training set, collected from the training domains $\mathcal{E}_{train}$, as $S = \bigcup_{e \in \mathcal{E}_{train}} S_e$, where $S_e$ is the sample set from domain $e$.

## B.1. Sample Size Within Domains - General Case

In this subsection we assume $\mathcal{L}(h, e) = \mathbb{E}_{(x,y) \sim P_e}[l(h(x), y)]$ for some cost function $l$.

**Theorem B.1.** *Let $\mathcal{H}$, $\mathcal{L}$, $\gamma$ follow the definitions from theorem 3.4 . Assume that:*

1. *$\mathcal{H}$ has the uniform-convergence property with respect to $\mathcal{L}, \gamma$ according to definition 3.3 with sample size $m(\delta, \epsilon)$.*

2. *$\mathcal{H}$ has also the uniform-convergence property according to the classical definition (in a single domain setting) with sample size $n(\delta, \epsilon)$.*

*For each $\epsilon_1, \epsilon_2, \delta > 0$, if $|\mathcal{E}_{train}| \geq m(\frac{\delta}{2}, \epsilon_2) := m$, and $\forall e \in \mathcal{E}_{train} \; |S_e| \geq n(\frac{\delta}{2m}, \epsilon_1) := n$, than with probability higher than $1 - \delta$, and regardless of the distribution $D$ over $\mathcal{E}$ and all the distributions $P_e$ for $e \in \mathcal{E}_{train}$, it holds that:*

$$\forall h \in \mathcal{H} \quad [\forall_{e \in \mathcal{E}_{train}} \frac{1}{|S_e|} \sum_{i \in S_e} l(h(X_i), y_i) \leq \gamma] \implies \mathbb{E}_{e \sim D}[\mathbb{1}_{\mathcal{L},\gamma+\epsilon_1}(h, e)] \leq \epsilon_2.$$

*Proof.* let $\epsilon_1, \epsilon_2, \delta > 0$ be positive numbers, and assume $S = \bigcup_{e \in \mathcal{E}_{train}} S_e$ such that $|\mathcal{E}_{train}| \geq m$ and for each $e \in \mathcal{E}_{train}$ $|S_e| \geq n$.

From the uniform-convergence of $\mathcal{H}$ in the classical, single-domain, sense, we know for each domain $e \in \mathcal{E}_{train}$ that with probability at least $1 - \frac{\delta}{2m}$ it holds that:

$$\forall h \in \mathcal{H} \quad \mathcal{L}(h, e) \leq \frac{1}{|S_e|} \sum_{i \in S_e} l(h(X_i), y_i) + \epsilon_1$$

And so,

$$\forall h \in \mathcal{H} \quad \frac{1}{|S_e|} \sum_{i \in S_e} l(h(X_i), y_i) \leq \gamma \implies \mathcal{L}(h, e) \leq \gamma + \epsilon_1 \implies \mathbb{1}_{\mathcal{L},\gamma+\epsilon_1}(h, e) = 0$$

This is true for each $e \in \mathcal{E}_{train}$, therefore with probability at leat $1 - \frac{\delta}{2}$ it is true for all training domains at once:

$$\forall h \in \mathcal{H} \quad [\forall_{e \in \mathcal{E}_{train}} \frac{1}{|S_e|} \sum_{i \in S_e} l(h(X_i), y_i) \leq \gamma] \implies \frac{1}{|\mathcal{E}_{train}|} \sum_{e \in \mathcal{E}_{train}} \mathbb{1}_{\mathcal{L},\gamma+\epsilon_1}(h, e) = 0$$

From the uniform-convergence property of $\mathcal{H}$ in the OOD sense, we know that with probability at least $1 - \frac{\delta}{2}$ it holds that

$$\forall h \in \mathcal{H} \quad \left| \mathbb{E}_{e \sim D}[\mathbb{1}_{\mathcal{L},\gamma+\epsilon_1}(h, e)] - \frac{1}{|\mathcal{E}_{train}|} \sum_{e \in \mathcal{E}_{train}} \mathbb{1}_{\mathcal{L},\gamma+\epsilon_1}(h, e) \right| \leq \epsilon_2.$$

And it the case of $\frac{1}{|\mathcal{E}_{train}|} \sum_{e \in \mathcal{E}_{train}} \mathbb{1}_{\mathcal{L},\gamma+\epsilon_1}(h, e) = 0$ we get:

$$\forall h \in \mathcal{H} \quad \left| \mathbb{E}_{e \sim D}[\mathbb{1}_{\mathcal{L},\gamma+\epsilon_1}(h, e)] \right| \leq \epsilon_2.$$

Overall, we have shown that with probability at least $1 - \delta$:

$$\forall h \in \mathcal{H} \quad \forall_{e \in \mathcal{E}_{train}} \frac{1}{|S_e|} \sum_{i \in S_e} l(h(X_i), y_i) \leq \gamma \implies \left| \mathbb{E}_{e \sim D}[\mathbb{1}_{\mathcal{L},\gamma+\epsilon_1}(h, e)] \right| \leq \epsilon_2.$$

$\square$

## B.2. Sample Size Within Domains - Recall Loss

Now we focus on $\mathcal{L}_{recall}$. We start by reminding its definition:

$$\mathcal{L}_{recall}(h,e) = \max_{y \in \mathcal{Y}} P_e[y \notin h(X)|Y = y].$$

Now we also assume that $\mathcal{H}$ is an hypothesis set of set-prediction hypotheses, where each $h \in \mathcal{H}$ can be decomposed to $|\mathcal{Y}|$ binary classifiers $h_y$ as presented in the paper. We assume also that the $h_y$ binary classifiers come from some $\mathcal{H}^*$ hypothesis set.

The following result differs from that of section B.1 mainly because $\mathcal{L}_{recall}$ is not an expectation over some other loss $l$. The result we derive for $\mathcal{L}_{recall}$ is almost the same as in the previous section, with only one change: Instead of requiring a sample size of $n(\frac{\delta}{2m}, \epsilon_1)$ at each training domain, we need to require a sample size of $n(\frac{\delta}{2m|\mathcal{Y}|}, \epsilon_1)$ for each training domain and each label $y \in \mathcal{Y}$. For completeness we present here the full result for $\mathcal{L}_{recall}$ and provide a full proof for it.

We add a single notation to this section:

$$S_{e,y} = \{i \in S_e \ : \ Y_i = y\}$$

**Theorem B.2.** *Let $\mathcal{H}$, $\mathcal{L}_{recall}$, $\gamma$ follow the definitions from the paper. Assume that:*

1. *$\mathcal{H}$ has the uniform-convergence property with respect to $\mathcal{L}_{recall}, \gamma$ according to definition 3.3 with sample size $m(\delta, \epsilon)$.*

2. *$\mathcal{H}^*$ has the uniform-convergence property according to the classical definition (in a single domain setting) with sample size $n(\delta, \epsilon)$.*

*For each $\epsilon_1, \epsilon_2, \delta > 0$, if $|\mathcal{E}_{train}| \geq m(\frac{\delta}{2}, \epsilon_2) := m$, and $\forall e \in \mathcal{E}_{train} \forall y \in \mathcal{Y} \ |S_{e,y}| \geq n(\frac{\delta}{2m|\mathcal{Y}|}, \epsilon_1) := n$, than with probability higher than $1 - \delta$, and regardless of the distribution $D$ over $\mathcal{E}$ and all the distributions $P_e$ for $e \in \mathcal{E}_{train}$, it holds that:*

$$\forall h \in \mathcal{H} \quad \left[ \forall_{e \in \mathcal{E}_{train}} \forall_{y \in \mathcal{Y}} \frac{1}{|S_{e,y}|} \sum_{i \in S_{e,y}} [1 - h_y(X_i)] \leq \gamma \right] \implies \underset{e \sim D}{\mathbb{E}}[\mathbb{1}_{\mathcal{L}_{recall}, \gamma + \epsilon_1}(h, e)] \leq \epsilon_2.$$

*Proof.* The main difference in this proof will be to show that with high probability

$$\forall h \in \mathcal{H} \quad \mathcal{L}_{recall}(h, e) \leq \max_{y \in \mathcal{Y}} \frac{1}{|S_{e,y}|} \sum_{i \in S_{e,y}} [1 - h_y(X_i)] + \epsilon_1$$

For completness we provide below the full proof.

Let $\epsilon_1, \epsilon_2, \delta > 0$ be positive numbers, and assume $S = \bigcup_{e \in \mathcal{E}_{train}} S_e$ such that $|\mathcal{E}_{train}| \geq m$ and $\forall e \in \mathcal{E}_{train} \forall y \in \mathcal{Y} \ |S_{e,y}| \geq n$.

From the uniform-convergence of $\mathcal{H}^*$ in the classical, single-domain, sense, we know for each domain $e \in \mathcal{E}_{train}$ and for each $y \in \mathcal{Y}$ that with probability at least $1 - \frac{\delta}{2m|\mathcal{Y}|}$ it holds that:

$$\forall h \in \mathcal{H}^* \quad P_e[h(X) \neq 1|Y = y] = \mathbb{E}_e[1 - h(X)|Y = y] \leq \frac{1}{|S_{e,y}|} \sum_{i \in S_{e,y}} [1 - h(X_i)] + \epsilon_1$$

This is true for each $y$ separately, so with probability at least $1 - \frac{\delta}{2m}$ this is true for all $y$ at once:

$$\forall h \in \mathcal{H}^* \ \forall y \in \mathcal{Y} \quad P_e[h(X) \neq 1|Y = y] \leq \frac{1}{|S_{e,y}|} \sum_{i \in S_{e,y}} [1 - h(X_i)] + \epsilon_1$$

We note that the above is true for hypotheses from $\mathcal{H}^*$, which are binary classifiers.

Now, for a given $h \in \mathcal{H}$, which is a set-valued predictor, let

$$y' = \arg\max_{y \in \mathcal{Y}} P_e[y \notin h(X)|Y = y].$$

Then, with probability at least $1 - \frac{\delta}{2m}$ it holds that:

$$\mathcal{L}_{recall}(h, e) = \max_{y \in \mathcal{Y}} P_e[y \notin h(X)|Y = y] = P_e[h_{y'}(X) \neq 1|Y = y'] \leq$$

$$\leq \frac{1}{|S_{e,y'}|} \sum_{i \in S_{e,y'}} [1 - h_{y'}(X_i)] + \epsilon_1 \leq \max_{y \in \mathcal{Y}} \frac{1}{|S_{e,y}|} \sum_{i \in S_{e,y}} [1 - h_y(X_i)] + \epsilon_1$$

Overall, we have shown that with probability at least $1 - \frac{\delta}{2m}$:

$$\forall h \in \mathcal{H} \quad \mathcal{L}_{recall}(h, e) \leq \max_{y \in \mathcal{Y}} \frac{1}{|S_{e,y}|} \sum_{i \in S_{e,y}} [1 - h_y(X_i)] + \epsilon_1$$

And so,

$$\forall h \in \mathcal{H} \quad \max_{y \in \mathcal{Y}} \frac{1}{|S_{e,y}|} \sum_{i \in S_{e,y}} [1 - h_y(X_i)] \leq \gamma \implies \mathcal{L}_{recall}(h, e) \leq \gamma + \epsilon_1 \implies$$

$$\implies \mathbb{1}_{\mathcal{L}_{recall}, \gamma + \epsilon_1}(h, e) = 0$$

This is true for each $e \in \mathcal{E}_{train}$, therefore with probability at leat $1 - \frac{\delta}{2}$ it is true for all training domains at once:

$$\forall h \in \mathcal{H} \quad [\forall_{e \in \mathcal{E}_{train}} \max_{y \in \mathcal{Y}} \frac{1}{|S_{e,y}|} \sum_{i \in S_{e,y}} [1 - h_y(X_i)] \leq \gamma] \implies \frac{1}{|\mathcal{E}_{train}|} \sum_{e \in \mathcal{E}_{train}} \mathbb{1}_{\mathcal{L}_{recall}, \gamma + \epsilon_1}(h, e) = 0$$

From the uniform-convergence property of $\mathcal{H}$ in the OOD sense, we know that with probability at least $1 - \frac{\delta}{2}$ it holds that

$$\forall h \in \mathcal{H} \quad \left| \mathbb{E}_{e \sim D}[\mathbb{1}_{\mathcal{L}_{recall}, \gamma + \epsilon_1}(h, e)] - \frac{1}{|\mathcal{E}_{train}|} \sum_{e \in \mathcal{E}_{train}} \mathbb{1}_{\mathcal{L}_{recall}, \gamma + \epsilon_1}(h, e) \right| \leq \epsilon_2.$$

And it the case of $\frac{1}{|\mathcal{E}_{train}|} \sum_{e \in \mathcal{E}_{train}} \mathbb{1}_{\mathcal{L}_{recall}, \gamma + \epsilon_1}(h, e) = 0$ we get:

$$\forall h \in \mathcal{H} \quad \left| \mathbb{E}_{e \sim D}[\mathbb{1}_{\mathcal{L}_{recall}, \gamma + \epsilon_1}(h, e)] \right| \leq \epsilon_2.$$

Overall, we have shown that with probability at least $1 - \delta$:

$$\forall h \in \mathcal{H} \quad \forall_{e \in \mathcal{E}_{train}} \forall_{y \in \mathcal{Y}} \frac{1}{|S_{e,y}|} \sum_{i \in S_{e,y}} [1 - h_y(X_i)] \leq \gamma \implies \mathbb{E}_{e \sim D}[\mathbb{1}_{\mathcal{L}_{recall}, \gamma + \epsilon_1}(h, e)] \leq \epsilon_2.$$

$\square$

### B.3. Sample Size Within Domains - Recall Loss With Linear Hypotheses

finally, we show the sample complexity for $\mathcal{L}_{recall}$ when $\mathcal{H}$ is the set of linear hypotheses.

**Theorem B.3.** *Let $\mathcal{H}$ be the hypothesis set of linesr set predictors in $\mathbb{R}^d$. Assume domains are restricted to being Conditionally Gaussian as described in theorem 3.7.*
*For each $\epsilon_1, \epsilon_2, \delta > 0$, if $|\mathcal{E}_{train}| \geq \Theta(\frac{|\mathcal{Y}|^2(d + \log(2|\mathcal{Y}|/\delta))}{\epsilon_2^2}) := m$, and $\forall e \in \mathcal{E}_{train} \forall y \in \mathcal{Y} \ |S_{e,y}| \geq \Theta(\frac{d + \log(2m|\mathcal{Y}|/\delta))}{\epsilon_1^2})$, than with probability higher than $1 - \delta$, and regardless of the distribution $D$ over $\mathcal{E}$ and all the distributions $P_e$ for $e \in \mathcal{E}_{train}$, it holds that:*

$$\forall h \in \mathcal{H} \quad [\forall_{e \in \mathcal{E}_{train}} \forall_{y \in \mathcal{Y}} \frac{1}{|S_{e,y}|} \sum_{i \in S_{e,y}} [1 - h_y(X_i)] \leq \gamma] \implies \mathbb{E}_{e \sim D}[\mathbb{1}_{\mathcal{L}_{recall}, \gamma + \epsilon_1}(h, e)] \leq \epsilon_2.$$

*Proof.* Let $\epsilon_1, \epsilon_2, \delta > 0$ be positive numbers, and assume $S = \bigcup_{e \in \mathcal{E}_{train}} S_e$ such that $|\mathcal{E}_{train}| \geq \Theta(\frac{|\mathcal{Y}|^2 (d + log(2|\mathcal{Y}|/\delta))}{\epsilon_2^2})$ and $\forall e \in \mathcal{E}_{train} \forall y \in \mathcal{Y} \; |S_{e,y}| \geq \Theta(\frac{d + log(2m|\mathcal{Y}|/\delta))}{\epsilon_1^2})$.

In the classical, single-domain context, linear hypotheses have $VC - dim = d + 1$, and they hold the uniform-convergence property with $n(\delta, \epsilon) = \Theta(\frac{d + log(1/\delta))}{\epsilon^2})$ (Shalev-Shwartz & Ben-David, 2014).

It holds that:

$$\forall e \in \mathcal{E}_{train} \forall y \in \mathcal{Y} \quad |S_{e,y}| \geq \Theta(\frac{d + log(2m|\mathcal{Y}|/\delta))}{\epsilon_1^2}) \implies |S_{e,y}| \geq n(\frac{\delta}{2m|\mathcal{Y}|}, \epsilon_1)$$

Following the exact same steps from the proof of the previous section, we can derive that with probability at leat $1 - \frac{\delta}{2}$:

$$\forall h \in \mathcal{H} \quad [\forall_{e \in \mathcal{E}_{train}} \max_{y \in \mathcal{Y}} \frac{1}{|S_{e,y}|} \sum_{i \in S_{e,y}} [1 - h_y(X_i)] \leq \gamma] \implies \forall e \in \mathcal{E}_{train} \mathbb{1}_{\mathcal{L}_{recall}, \gamma + \epsilon_1}(h, e) = 0$$

Now, from theorem 3.7 we know that with probability at leat $1 - \frac{\delta}{2}$:

$$\forall h \in \mathcal{H} \quad \forall e \in \mathcal{E}_{train} \mathbb{1}_{\mathcal{L}_{recall}, \gamma + \epsilon_1}(h, e) = 0 \implies \mathbb{E}_{e \sim D}[\mathbb{1}_{\mathcal{L}_{recall}, \gamma + \epsilon_1}(h, e)] \leq \epsilon_2.$$

Together, we get that with probability at least $1 - \delta$:

$$\forall h \in \mathcal{H} \quad [\forall_{e \in \mathcal{E}_{train}} \max_{y \in \mathcal{Y}} \frac{1}{|S_{e,y}|} \sum_{i \in S_{e,y}} [1 - h_y(X_i)] \leq \gamma] \implies \mathbb{E}_{e \sim D}[\mathbb{1}_{\mathcal{L}_{recall}, \gamma + \epsilon_1}(h, e)] \leq \epsilon_2.$$

$\square$

## C. Developing set-size optimization method

In section 4.1 we presented our learning objective as a constrained optimization problem for each $y \in \mathcal{Y}$ separately:

$$\min_{h \in \mathcal{H}} \frac{1}{N} \sum_{i \in S} h_y(X_i)$$
$$\text{s.t} \quad \forall e \in \mathcal{E}_{train} \frac{1}{|G_{e,y}|} \sum_{i \in G_{e,y}} \mathbb{1}[h_y(X_i) = 0] \leq \gamma.$$

This problem can also be written as:

$$\min_{h \in \mathcal{H}} \frac{1}{N} \sum_{i=1}^{N} h_y(X_i)$$
$$\text{s.t} \quad \forall e \in \mathcal{E}_{train} \frac{1}{|G_{e,y}|} \sum_{i \in G_{e,y}} (1 - h_y(X_i)) \leq \gamma. \tag{3}$$

We now investigate the problem and further develop it, until we get a lagrangian that approximates the original problem, and which is possible to optimize using common methods.

### C.1. Relaxed Problem

In order to leverage common gradient-based optimization methods, we parameterize our predictors as mentioned in section 4.1 and present slack variables to form the following relaxed problem:

$$\min_{\theta} \quad \frac{1}{N} \sum_i \xi_i$$

$$\text{s.t} \quad \forall i \quad h_y^{\theta}(X) \leq -1 + \xi_i$$

$$\forall i \in G_y \quad h_y^{\theta}(X) \geq 1 - \zeta_i$$

$$\forall e \in E \quad \frac{1}{G_{e,y}} \sum_{i \in G_{e,y}} \zeta_i \leq \epsilon \tag{4}$$

$$\forall i \quad \zeta_i \geq 0$$

$$\forall i \quad \xi_i \geq 0$$

**Theorem C.1.** *The objective of the relaxed problem (4) is a surrogate for the objective of the original problem (3)*

*Proof.* From the first constraint of (4) we have for each $\theta$

$$\xi_i \geq 1 \iff h_y^{\theta}(X_i) \geq 0 \iff h_y(x_i) = 1$$

$$0 \leq \xi_i < 1 \iff h_y^{\theta}(X_i) < 0 \iff h_y(x_i) = 0$$

Thus, $\xi_i \geq h_y(X_i)$, making $\sum_i \xi_i$ a surrogate loss for $\sum_i h_y(X_i)$.

$\square$

**Theorem C.2.** *The recall constraint of the relaxed problem (4) is a surrogate for the recall constraint of the original problem 3)*

*Proof.* From the second constraint of (4) we have for $i \in G_y$:

$$\zeta_i \geq 1 \iff h_y^{\theta}(X_i) \leq 0 \iff h_y(x_i) = 0 \iff 1 - h_Y(X_i) = 1$$

$$0 \leq \zeta_i < 1 \iff h_y^{\theta}(X_i) > 0 \iff h_y(x_i) = 1 \iff 1 - h_Y(X_i) = 0$$

Thus, $\zeta_i \geq 1 - h_Y(X_i)$ for $i \in G_y$, making $\frac{1}{|G_{e,y}|} \sum_{i \in G_{e,1}} \zeta_i$ a surrogate loss for $\frac{1}{|G_{e,y}|} \sum_{i \in G_{e,y}} 1 - h_Y(X_i)$. $\square$

### C.2. Lagrangian Form

Solving the relaxed problem for $\xi_i$ and $\zeta_i$ we get:

$$\xi_i = max[0, 1 + h_y^{\theta}(X_i)]$$

$$\zeta_i = max[0, 1 - h_y^{\theta}(X_i)]$$

Where the expression for $\zeta_i$ is one of the optimal solutions for it (there are additional optimal solutions). Substituting these into the optimization problem (4) yields:

$$\min_{\theta} \quad \sum_i max[0, 1 + h_y^{\theta}(X_i)]$$

$$\text{s.t} \quad \forall e \in E \quad \frac{1}{|G_{e,1}|} \sum_{i \in G_{e,y}} max[0, 1 - h_y^{\theta}(X_i)] \leq \gamma \tag{5}$$

This problem can be optimized using common methods like Stochastic Gradient Descent (SGD) by defining the lagrangian:

$$L(\theta, C) = \sum_i max[0, 1 + h_y^{\theta}(X_i)] + \sum_{e \in E_{train}} C_{e,y} \left( \frac{1}{|G_{e,y}|} \sum_{i \in G_{e,y}} max[0, 1 - h_y^{\theta}(X_i)] - \gamma \right)$$

Solving $\min_\theta \max_C L(\theta, C)$ will give $\theta$ that solves the constrained optimization problem 5. common approach is to use gradient descent on $\theta$, and gradient ascent on C. This is being done by SET-COVER, as described in section 4.1

## D. SET-COVER Pseudo Code

**Algorithm 1** SET-COVER

Initialize $\theta, C$
**for** i from 1 to NUM EPOCHS **do**
    **for** $b$ in batches **do**
        Call COMPUTE $L_y(\theta, C)$
        $L(\theta, C) = \sum_{y \in \mathcal{Y}} L_y(\theta, C)$
        perform GD step for $\theta$ with respect to $L(\theta, C)$
        **IF** i % C UPDATE FREQUENCY == 0 **do**
            Call UPDATE_C
    **end for**
    Call UPDATE_C
**end for**
**Subroutine:** COMPUTE $L_y(\theta, C)$

$$L_y(\theta, C) = \sum_{i \in b} 1_{Y_i \neq y} \max\{0, 1 + h_y^\theta(X_i)\} +$$
$$1_{Y_i = y} \cdot C_{e_i, y} \cdot \max\{0, 1 - h_y^\theta(X_i)\}$$

**Subroutine:** UPDATE_C
    **for** $e \in \mathcal{E}_{train}$ **do**
        **for** $y \in \mathcal{Y}$ **do**
            coverage $\leftarrow \frac{1}{|G_{e,y}|} \sum_{i \in G_{e,y}} \mathbb{1}[h_y^\theta(X_i) > 0]$
            $\nu \leftarrow 1 - (\text{coverage}_{e,y} - (1 - \gamma))$
            $s \leftarrow 2$    if $\nu > 1$    else 1
            $C_{e,y} \leftarrow C_{e,y} \cdot s \cdot \nu$
        **end for**
    **end for**

## E. Experiments

### E.1. Experiments Hyper-Parameters

Table 2. hyper-parameters used for our experiments

| Hyper-Parameter | Camelyon | Fmow | Iwildcam | Amazon | Synthetic |
|---|---|---|---|---|---|
| Batch Size | 128 | 64 | 64 | 128 | 128 |
| Learning Rate | 0.001 | 0.001 | 0.001 | 0.001 | 0.001 |
| Number of Epochs | 5 | 5 | 5 | 5 | 30 |
| Number of train domains | 20 | 20 | 80 | 500 | 25 |
| Number of test domains | 20 | 18 | 40 | 100 | 25 |
| Max train domain size | 6,000 | 4,500 | 3,000 | 1,000 | 2,000 |
| Max test domain size | 2,000 | 3,000 | 1,000 | 1,000 | 1,000 |
| **Relevant for SET-COVER:** | | | | | |
| Initial C value | 5 | 5 | 5 | 5 | 5 |
| Frequency of C values update | 500 | 500 | 500 | 500 | 500 |

Table 3. Hidden dimension of 2-layer MLP (in the relevant experiments)

| | Synthetic, d=10 | Synthetic, d=50 | Amazon |
|---|---|---|---|
| Hidden dim | 5 | 25 | 10 |

## E.2. Synthetic Data Generarion Process Parameters

We set $\Sigma$ to be a diagonal covariance matrix with $\sigma$ value on the diagonal. $\sigma$ is detailed in table 4 below.

Also, we set $\nu \in \mathbb{R}^d$ to be a concatenation of two vectors $\nu_1, \nu_2 \in \mathbb{R}^{0.5d}$, i.e $\nu = (\nu_1, \nu_2)$. $\nu_1, \nu_2$ are detailed in table 4 below.

*Table 4.* Synthetatic data generation parameters.

| dimension d | $u_{low}$ | $u_{high}$ | $\mu \in \mathbb{R}^d$ | $\nu_1 \in \mathbb{R}^{0.5d}$ | $\nu_2 \in \mathbb{R}^{0.5d}$ | $\sigma$ |
|---|---|---|---|---|---|---|
| $d = 10$ | -0.5 | 0.5 | (0.1, ... , 0.1) | (1, ... , 1) | (-1, ... , -1) | 0.2 |
| $d = 50$ | -0.3 | 0.3 | (0.05, ... , 0.05) | (1, ... , 1) | (-1, ... , -1) | 0.25 |

## E.3. Additional Synthetic Data Experiments

In Theorem 3.7 we have shown a theoretical generalization result, but under the limitation of shared covariance structure across domains (up to a scaling factor). Our results in the synthetic data experiment, presented in section 5.1 empirically support this result. In this section we want to test whether the generalization to new domains can hold also in DGPs where the covariance between domains does not share exactly same structure. To this end, we recall the DGP presented in section 5.1:

$$Z_e \sim U[u_{\text{low}}, u_{\text{high}}]$$
$$Y \sim Bernoulli(0.5)$$
$$X \sim Y(\mu + Z_e \nu) + N(0, \Sigma)$$

In the following experiment we change the covariance matrix to be domain-specific in the following way:

1. We sample for each domain a diagonal matrix, $D_e$, with diagonal values sampled from a normal distribtuion with $\mu = \sigma$ and $std = 0.05$ (this process generates std values, which are than squared to form the diagonal values of $D_e$). $\sigma$ values are the same as set in the original experiment from section 5.1.

$$D_e = diag([D_{e,1}^2, ..., D_{e,d}^2]$$

$$\forall 1 \leq i \leq d \quad D_{e,i} \sim N(\sigma, 0.05)$$

2. For each domain we sample uniformly a rotation matrix $Q_e$.

3. For each domain we set the covaraince matrix $\Sigma_e = Q_e^T D_e Q_e$

All other experiments' hyper-parameters are the same as the original experiment from section 5.1. The results are presented in Figure 3 and Table 5.

*Table 5.* OOD Performance on synthetic Datasets with Random Covariance

| Model | 10d | | | 50d | | |
|---|---|---|---|---|---|---|
| | Median Min Recall ↑ | Median Avg Size ↓ | Recall $\geq 90\%$ Pctg ↑ | Median Min Recall ↑ | Median Avg Size ↓ | Recall $\geq 90\%$ Pctg ↑ |
| ERM | 0.88 | 1.0 | 0.39 | 0.90 | 1.0 | 0.52 |
| CDF Pooling-(TrainC) | 0.87 | 1.02 | 0.30 | 0.86 | 0.98 | 0.19 |
| CDF Pooling-(CVC) | 0.87 | 1.04 | 0.32 | 0.86 | 1.04 | 0.27 |
| Robust-Conformal | 0.94 | 1.24 | 0.94 | 0.90 | 1.24 | 0.71 |
| SET-COVER | 0.94 | 1.23 | 0.92 | 0.91 | 1.18 | 0.68 |

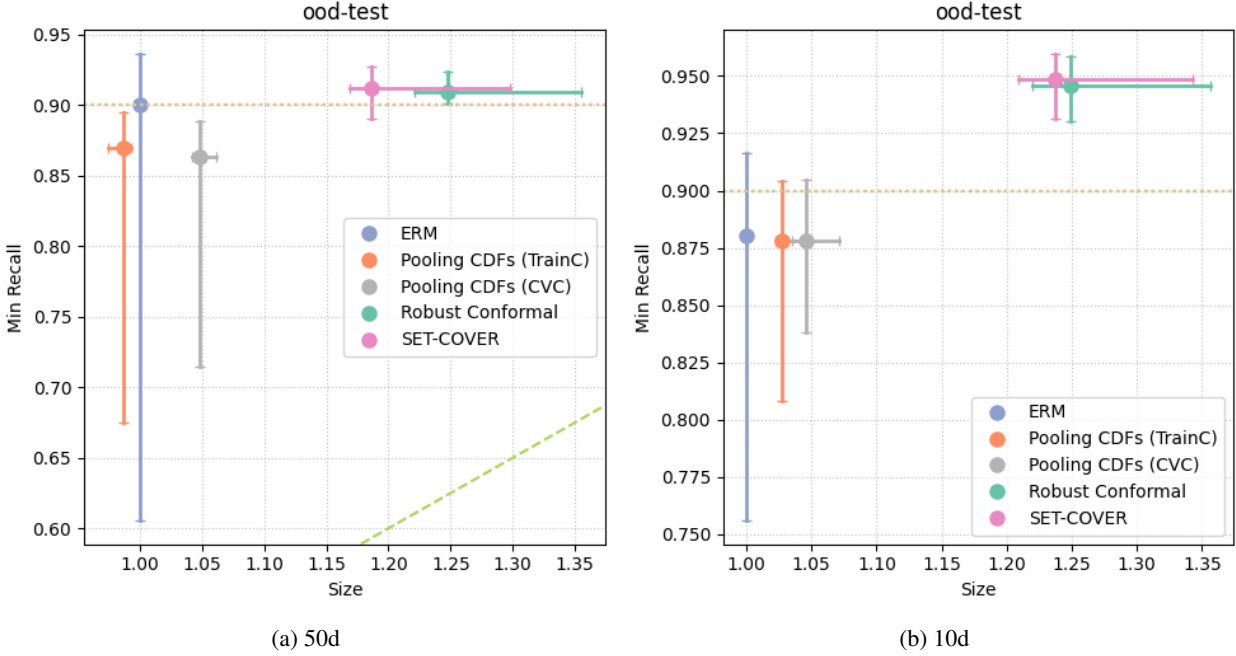

(a) 50d                                     (b) 10d

*Figure 3.* Min Recall distribution VS Mean Set Size distribution. **Blue** represents ERM model, **Orange** represents Pooling CDFs (TrainC), **Grey** represents Pooling CDFs (CVC), **Green** represents robust conformal, and **Pink** represents SET-COVER. The horizontal solid line represents the 90% recall target value.

### E.4. Exploring Different $\gamma$ Values

The $\gamma$ parameter sets the desired recall level, and is supposed to be set in practice by practitioners according to task requirement. In our main experiments, which are presented at the body of this work, we targeted at a 0.9 recall level, which is associated with $\gamma = 0.1$. In this subsection we present results also for targeted recall levels of 0.8 and 0.95.

E.4.1. **0.8 RECALL**

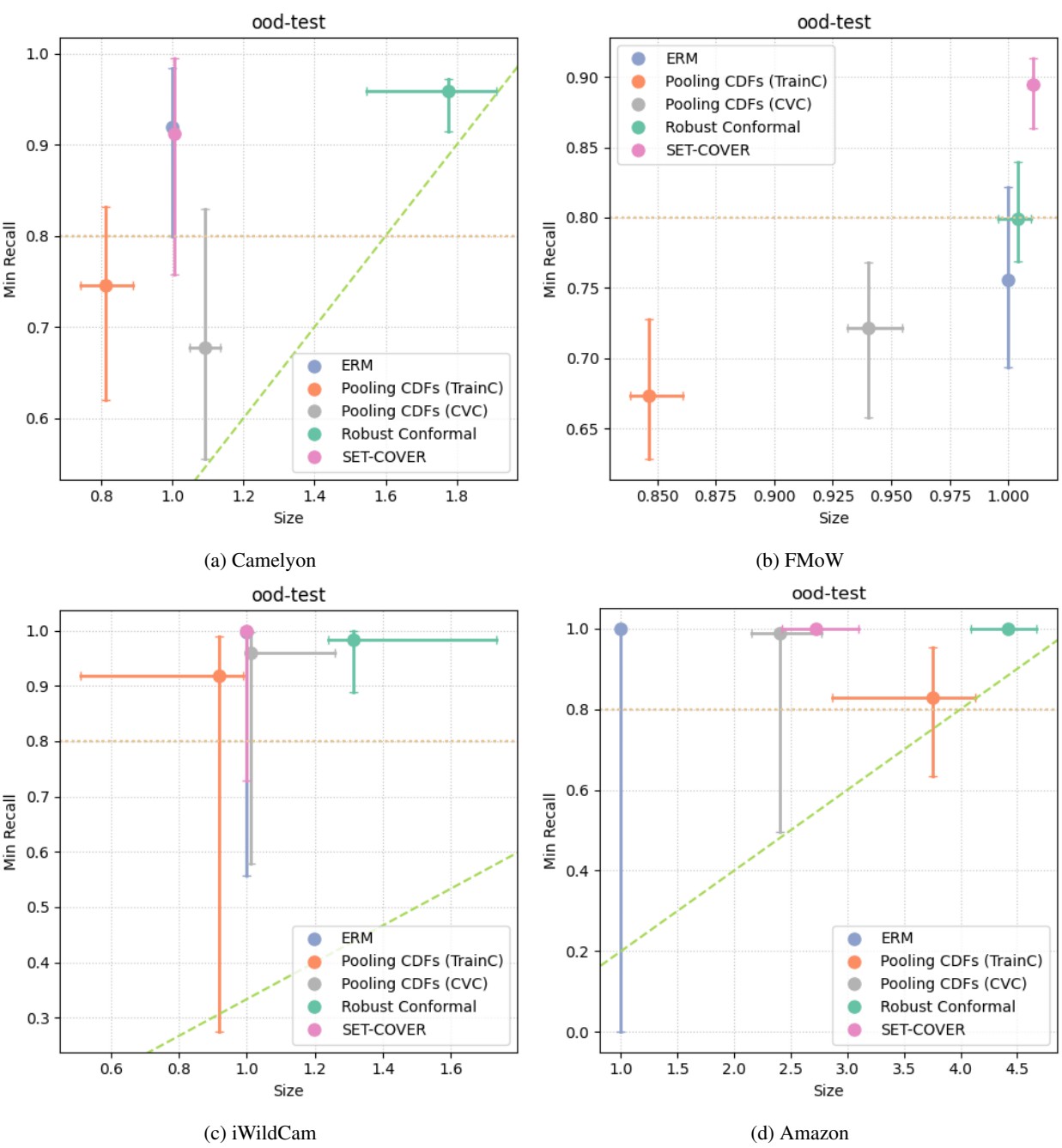

(a) Camelyon

(b) FMoW

(c) iWildCam

(d) Amazon

*Figure 4.* Results for recall target of 0.8 ($\gamma = 0.2$)

*Table 6.* Summary of OOD Results for recall level of 0.8 ($\gamma = 0.2$)

| Model | Camelyon | | | FMoW | | |
|---|---|---|---|---|---|---|
| | Median Min Recall ↑ | Median Avg Size ↓ | Recall ≥ 90% Pctg ↑ | Median Min Recall ↑ | Median Avg Size ↓ | Recall ≥ 90% Pctg ↑ |
| **ERM** | 0.91 | 1.0 | 0.75 | 0.75 | 1.0 | 0.42 |
| **CDF Pooling-(TrainC)** | 0.74 | 0.81 | 0.38 | 0.67 | 0.84 | 0.05 |
| **CDF Pooling-(CVC)** | 0.67 | 1.09 | 0.45 | 0.72 | 0.94 | 0.16 |
| **Robust Conformal** | 0.95 | 1.77 | 0.90 | 0.79 | 1.00 | 0.50 |
| **SET-COVER** | 0.91 | 1.00 | 0.71 | 0.89 | 1.01 | 0.94 |

| Model | iWildCam | | | Amazon | | |
|---|---|---|---|---|---|---|
| | Median Min Recall ↑ | Median Avg Size ↓ | Recall ≥ 90% Pctg ↑ | Median Min Recall ↑ | Median Avg Size ↓ | Recall ≥ 90% Pctg ↑ |
| **ERM** | 0.99 | 1.0 | 0.71 | 1.0 | 1.0 | 0.69 |
| **CDF Pooling-(TrainC)** | 0.91 | 0.91 | 0.60 | 0.83 | 3.75 | 0.56 |
| **CDF Pooling-(CVC)** | 0.95 | 1.01 | 0.70 | 0.99 | 2.40 | 0.70 |
| **Robust Conformal** | 0.98 | 1.31 | 0.76 | 1.0 | 4.41 | 1.0 |
| **SET-COVER** | 0.99 | 1.00 | 0.71 | 1.0 | 2.72 | 0.98 |

E.4.2. **0.95 RECALL**

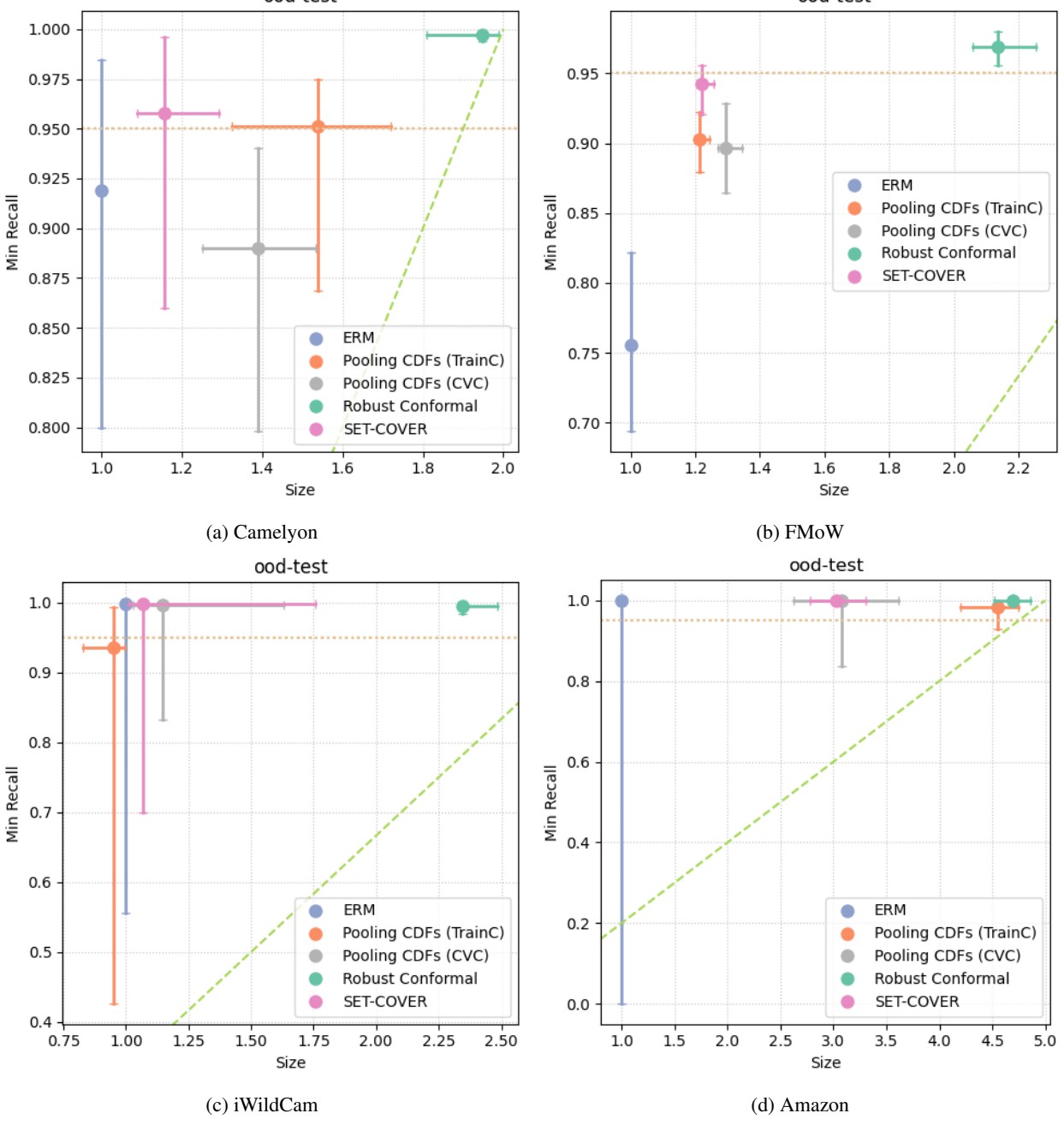

(a) Camelyon

(b) FMoW

(c) iWildCam

(d) Amazon

*Figure 5.* Results for recall target of 0.95 ($\gamma = 0.05$)

*Table 7.* Summary of OOD Results for recall level of 0.95 ($\gamma = 0.05$)

| Model | Camelyon | | | FMoW | | |
|---|---|---|---|---|---|---|
| | Median Min Recall ↑ | Median Avg Size ↓ | Recall ≥ 90% Pctg ↑ | Median Min Recall ↑ | Median Avg Size ↓ | Recall ≥ 90% Pctg ↑ |
| **ERM** | 0.91 | 1.0 | 0.48 | 0.75 | 1.0 | 0.03 |
| **CDF Pooling-(TrainC)** | 0.95 | 1.53 | 0.5 | 0.90 | 1.21 | 0.07 |
| **CDF Pooling-(CVC)** | 0.88 | 1.39 | 0.26 | 0.89 | 1.29 | 0.14 |
| **Robust Conformal** | 0.99 | 1.94 | 0.93 | 0.96 | 2.13 | 0.64 |
| **SET-COVER** | 0.95 | 1.15 | 0.65 | 0.94 | 1.22 | 0.53 |

| Model | iWildCam | | | Amazon | | |
|---|---|---|---|---|---|---|
| | Median Min Recall ↑ | Median Avg Size ↓ | Recall ≥ 90% Pctg ↑ | Median Min Recall ↑ | Median Avg Size ↓ | Recall ≥ 90% Pctg ↑ |
| **ERM** | 0.99 | 1.0 | 0.70 | 1.0 | 1.0 | 0.69 |
| **CDF Pooling-(TrainC)** | 0.93 | 0.94 | 0.45 | 0.98 | 4.54 | 0.69 |
| **CDF Pooling-(CVC)** | 0.99 | 1.14 | 0.72 | 1.0 | 3.07 | 0.72 |
| **Robust Conformal** | 0.99 | 2.34 | 0.85 | 1.0 | 4.69 | 1.0 |
| **SET-COVER** | 0.99 | 1.07 | 0.77 | 1.0 | 3.02 | 0.97 |

### E.4.3. RELATIONSHIP BETWEEN RECALL AND SET SIZE

In Figure 6 we illustrate how varying the recall target affects the resulting prediction set size, across methods and datasets. The results are based on the three $\gamma$ values presented earlier, corresponding to target recall levels of 0.8, 0.9, and 0.95. As expected, increasing the desired recall level generally leads to a corresponding increase in set size. This trade-off reflects the fundamental tension between coverage and specificity: higher recall necessitates larger prediction sets to ensure that the correct label is included. The trend is consistent across datasets and methods, although the magnitude of the size increase varies. Notably, SET-COVER tends to achieve high recall with relatively smaller increases in set size, indicating better efficiency in balancing recall and compactness. These observations reinforce the importance of selecting $\gamma$ (and the associated recall target) with awareness of the practical constraints and cost associated with larger prediction sets.

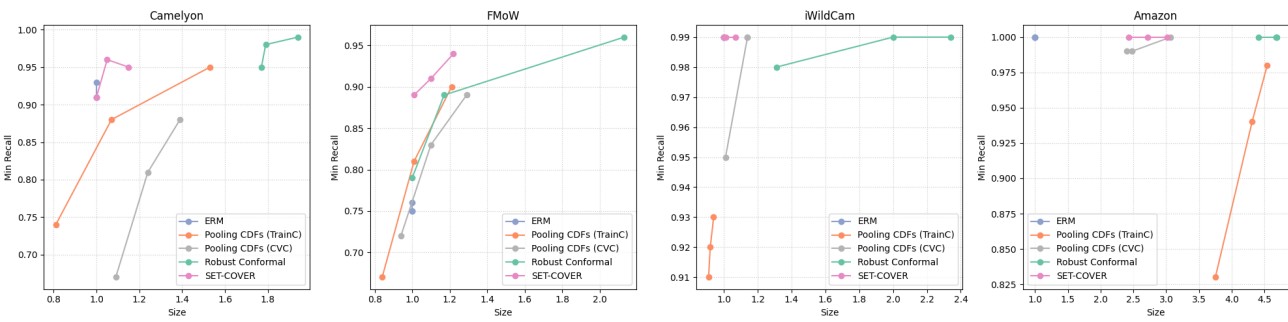

*Figure 6.* Relationship between recall and set size. Each curve corresponds to a method and shows results for three target recall levels ($\gamma \in \{0.2, 0.1, 0.05\}$, corresponding to target recalls of 0.8, 0.9, and 0.95). The Y-axis indicates the actual minimum recall achieved, while the X-axis shows the corresponding prediction set size.

### E.5. Experiments on Other OOD Baselines

For the main experiments of this work we chose the ERM method as the single-prediction basekine, due to its vast popularity in real-world applications, and its superior, or at least compatible performance in various OOD baselines (Koh et al., 2021; Gulrajani & Lopez-Paz, 2020). In the next section we compare SET-COVER to additional common OOD baselines. These include IRM (Arjovsky et al., 2019), VREx (Krueger et al., 2021), MMD (Li et al., 2018b), and CORAL (Sun & Saenko, 2016). We use the DomainBed (Gulrajani & Lopez-Paz, 2020) package to train these models.

The results show that common single-prediction baselines do not maintain the 90% min-recall target across most OOD domains. SET-COVER presents an advantage in getting the target min-recall level across unseen domains, suggesting that set-valued predictors may be a step in the right direction for robust OOD generalization.

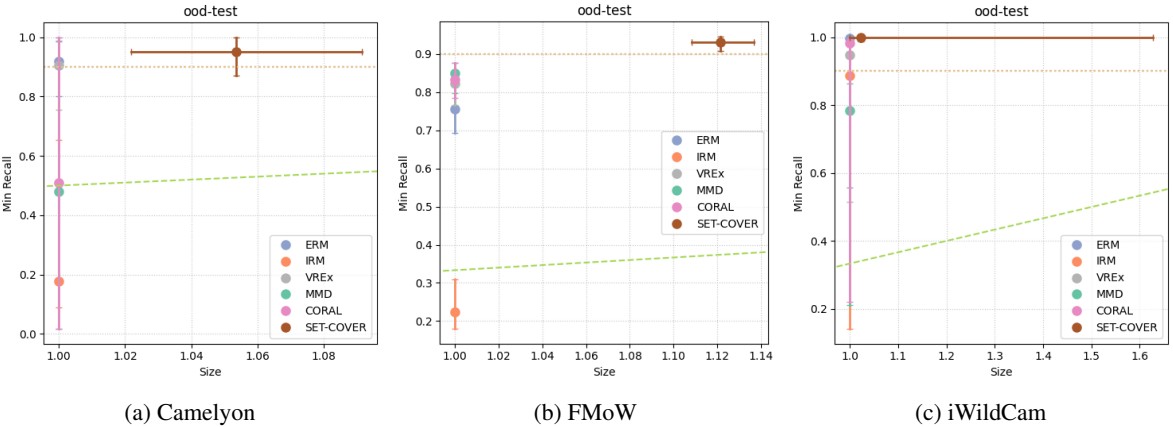

| (a) Camelyon | (b) FMoW | (c) iWildCam |

*Figure 7.* Each figure represents Min-Recall over Avg Set Size cross. y-axis represents min-recall, and x-axis represents average set size. Each cross shows the median and the 25th and 75th percentiles for both metrics across domain. The horizontal solid line represents the 90% recall target value, and dashed yellow diagonal line represents performance of a random predictor.

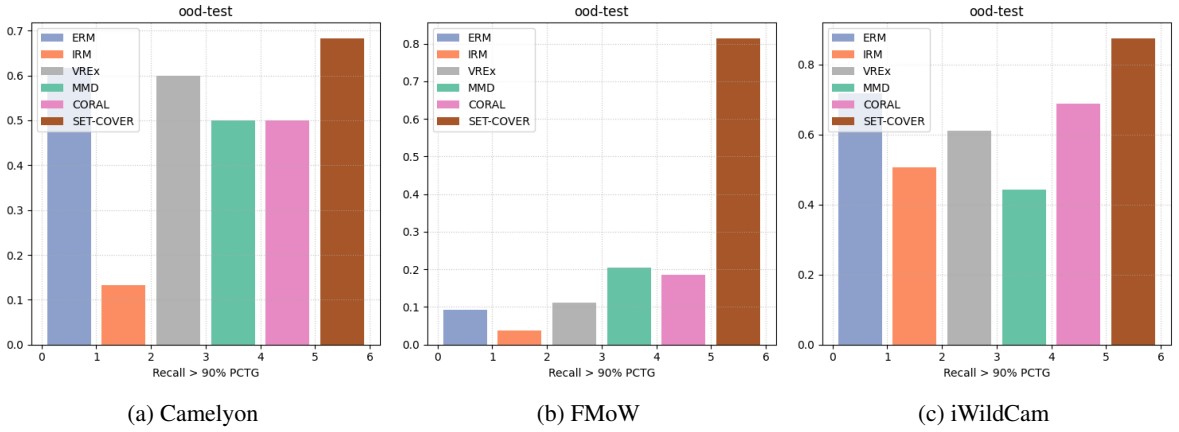

| (a) Camelyon | (b) FMoW | (c) iWildCam |

*Figure 8.* Percentage of OOD domains where the min-recall is higher than 90%. Each bar represents a different model.

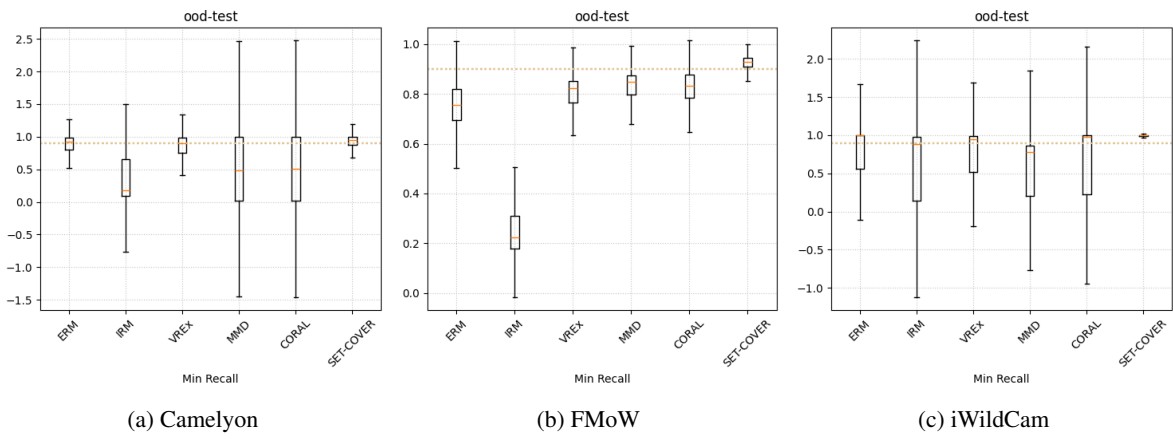

*Figure 9.* Boxplots represent the distribution of min-recall across OOD domains.

*Table 8.* Summary of OOD Results for different OOD baselines.

| Model | Camelyon | | | FMoW | | |
|---|---|---|---|---|---|---|
| | Median | Median | Recall $\geq 90\%$ | Median | Median | Recall $\geq 90\%$ |
| | Min Recall $\uparrow$ | Avg Size $\downarrow$ | Pctg $\uparrow$ | Min Recall $\uparrow$ | Avg Size $\downarrow$ | Pctg $\uparrow$ |
| **ERM** | 0.91 | 1.0 | 0.61 | 0.75 | 1.0 | 0.09 |
| **IRM** | 0.17 | 1.00 | 0.13 | 0.22 | 1.00 | 0.03 |
| **VREx** | 0.90 | 1.00 | 0.60 | 0.82 | 1.00 | 0.11 |
| **MMD** | 0.48 | 1.00 | 0.50 | 0.84 | 1.00 | 0.20 |
| **CORAL** | 0.50 | 1.00 | 0.50 | 0.83 | 1.00 | 0.18 |
| **SET-COVER** | 0.95 | 1.05 | 0.68 | 0.93 | 1.12 | 0.81 |

| Model | iWildCam | | |
|---|---|---|---|
| | Median | Median | Recall $\geq 90\%$ |
| | Min Recall $\uparrow$ | Avg Size $\downarrow$ | Pctg $\uparrow$ |
| **ERM** | 0.99 | 1.0 | 0.71 |
| **IRM** | 0.88 | 1.00 | 0.50 |
| **VREx** | 0.94 | 1.00 | 0.60 |
| **MMD** | 0.78 | 1.00 | 0.44 |
| **CORAL** | 0.98 | 1.00 | 0.68 |
| **SET-COVER** | 1.00 | 1.02 | 0.87 |

### E.6. Additional Results of Main Experiments

We present here additional results of the WILDS experiments presented in section 5.2.

### E.6.1. COVERAGE GENERALIZATION PLOT

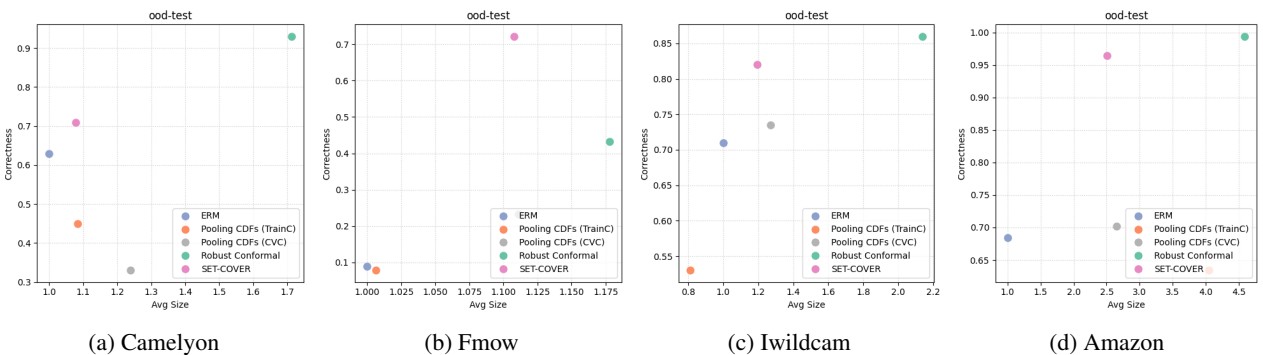

(a) Camelyon  (b) Fmow  (c) Iwildcam  (d) Amazon

*Figure 10.* Y-axis represents the percentage of domains with min-recall $\geq 90\%$. X-axis represents average set size. **Blue** represents ERM predictor, **Orange** represents Pooling CDFs (TrainC), **Grey** represents Pooling CDFs (CVC), **Green** represents robust conformal predictor, and **Pink** represents SET-COVER.

### E.6.2. SET-COVER LOSS VARIATIONS COMPARISSON

In section 4.1 we argue that SET-COVER should not penalize a correctly predicted label in the set-size loss term of the Lagrangian (the first addend of the Lagrangian). This led us to update the Lagrangian we optimize from

$$L_y(\theta, C) = \sum_{i \in S} \max\{0, 1 + h_y^\theta(X_i)\} + \\ \sum_{e \in E_{train}} \mathbb{1}_{i \in G_{e,y}} C_{e,y} \max\{0, 1 - h_y^\theta(X_i)\},$$

to

$$L_y(\theta, C) = \sum_{i \notin G_y} \max\{0, 1 + h_y^\theta(X_i)\} + \\ \sum_{e \in E_{train}} \mathbb{1}_{i \in G_{e,y}} C_{e,y} \max\{0, 1 - h_y^\theta(X_i)\}.$$

Here we compare the two variants. The first variant we call "Full Set Penalty", as it penalizes also the correct label in the set-size term of the Lagrangian. The second variant is called "Wrong Prediction Penalty", as it only penalizes the wrong labels in the prediction set. Figure 11 shows that the "Full Set Penalty" does not consistently improve set size, and in Camelyon and Iwildcam it even outputs somewhat larger sets. In addition, it does not lead to an improvement in coverage, and in Camelyon and Iwildcam it even leads to a moderate degradation in coverage.

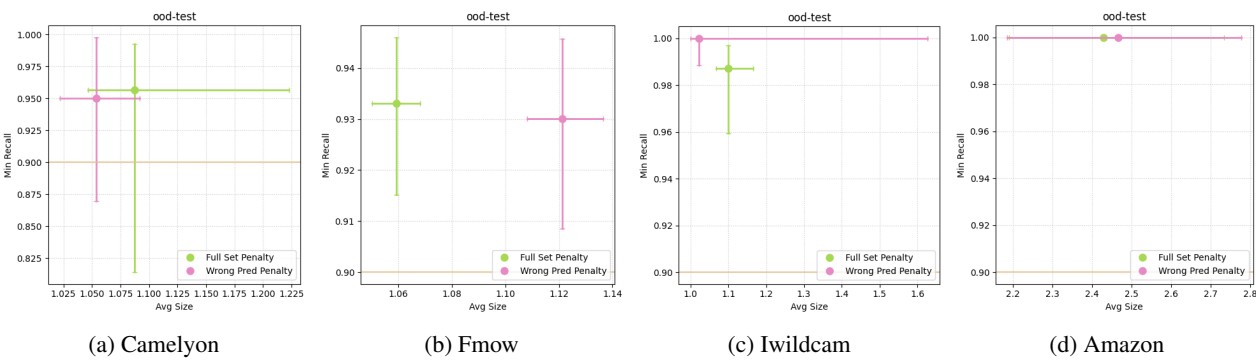

(a) Camelyon         (b) Fmow         (c) Iwildcam         (d) Amazon

*Figure 11.* Each figure represents Min-Recall over Avg Set Size cross. y-axis represents min-recall, and x-axis represents average set size. Each cross shows the median and the 25th and 75th percentiles for both metrics across domain. **Pink** represents SET-COVER as presented in the paper, i.e the "Wrong Prediction Penalty" variant. **Yellow** represents SET-COVER with loss penalizing entire set size, including correct labels, i.e the "Full Set Penalty".

### E.6.3. TRAINING TIME AND COMPUTATIONAL COST

We report the average training times (measured on a single NVIDIA GPU) for ERM and SET-COVER on each dataset in Table 9. SET-COVER incurs a moderate increase in training time (approximately 30% on average) compared to ERM, primarily due to the optimization of Lagrange multipliers ($C$ in our algorithm). Aside from this, SET-COVER shares similar computational requirements with ERM, relying on hinge-loss-based optimization without substantial architectural complexity.

*Table 9.* Average training times (in minutes) for ERM and SET-COVER across datasets.

| Dataset | ERM (min) | SET-COVER (min) |
|---|---|---|
| Camelyon | 98 | 133 |
| FMoW | 45 | 56 |
| iWildCam | 46 | 58 |
| Amazon | 12 | 15 |

The current implementation of SET-COVER can be further optimized by, for example, exploiting GPU parallelism more effectively. We anticipate that such improvements would significantly reduce the additional computational overhead.

Other set-valued predictors (Pooling CDFs, Robust Conformal) are trained in two stages where the first stage involves training an ERM classifier, which dominates the overall runtime. For simplicity, we approximate their training time by that of ERM. Additionally, some OOD baselines discussed in Appendix E.5 rely on DomainBed implementations, which apply various runtime optimizations, making direct training time comparisons inconsistent.

### E.6.4. CROSS PLOTS ON TRAINING DOMAINS

We present in Figure 12 the cross-plot that is presented in Figure 2, Section 5.2, but here we present it for Train set and In-Domain test set, alongside the OOD test set (which is also presented in Figure 2).

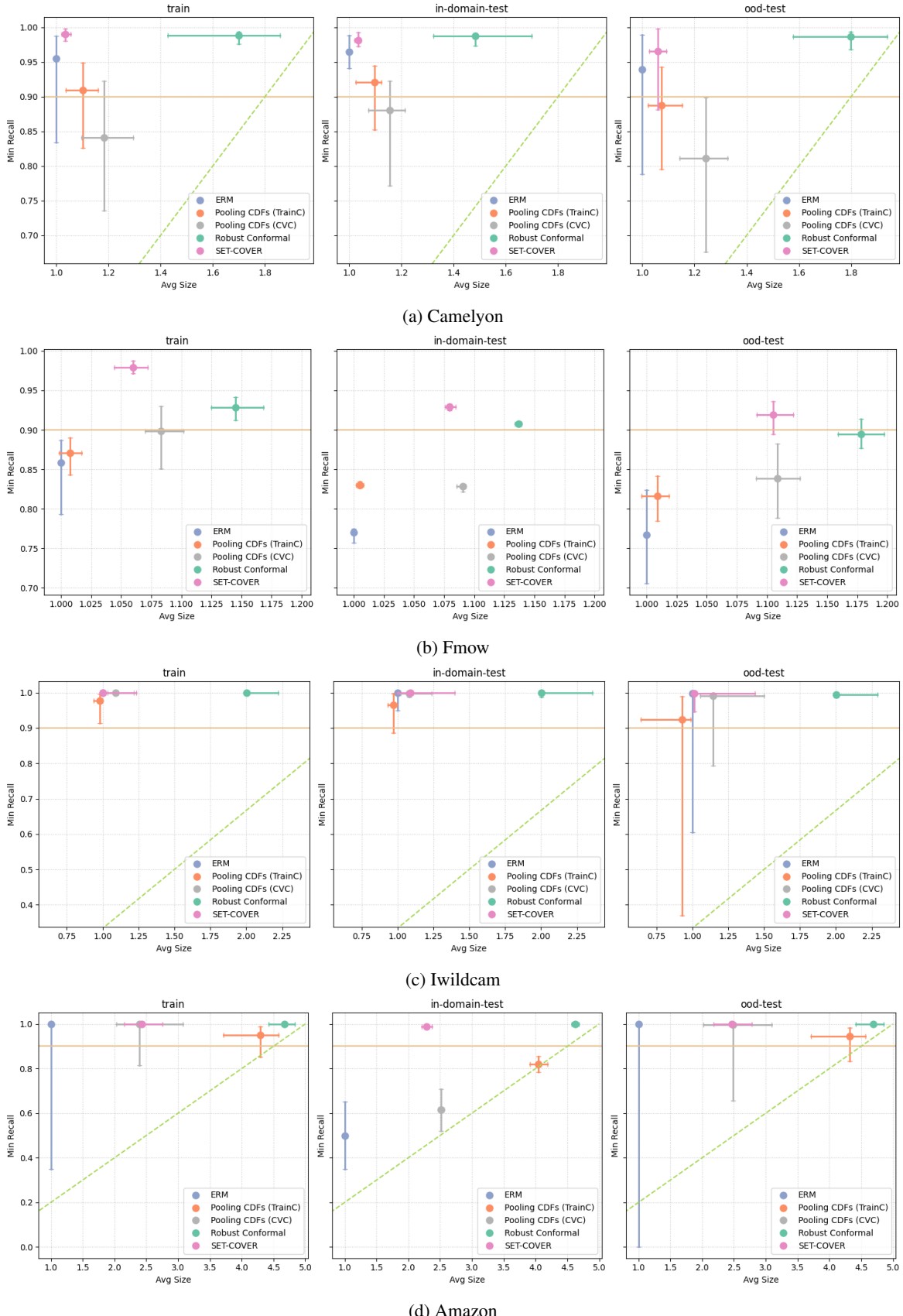

*Figure 12.* Cross Plots For All Data Sets

