# OpenReview forum: "Set Valued Predictions For Robust Domain Generalization"
_ICML.cc/2025/Conference — ICML 2025 poster_

### Official Review · Reviewer_Taio · 2025-03-12

**Overall Recommendation:** 4

**Summary:**

The paper introduces a set-valued prediction approach for robust Domain Generalization (DG). It argues that single-valued predictions limit robustness, proposing instead to predict sets of labels to achieve reliable coverage across unseen domains. The authors provide theoretical generalization bounds and introduce an optimization algorithm (SET-COVER) to minimize set size while maintaining performance guarantees. Experimental results on WILDS datasets show improvements over existing baselines in robustness and prediction set efficiency.

**Claims And Evidence:**

Overall, the paper's claims are largely supported, though some gaps remain.

The theoretical generalization results (VC-dimension-based bounds) rely on restrictive assumptions (e.g., conditional Gaussianity, identical covariance structures across domains.

**Essential References Not Discussed:**

None

**Experimental Designs Or Analyses:**

Yes, I reviewed the experimental designs, especially those involving real-world datasets from the WILDS benchmark.

One notable issue is that the paper's main experiments primarily emphasize recall and prediction set size but do not sufficiently analyze trade-offs such as computational cost, calibration robustness, or practical usability of large prediction sets.

**Methods And Evaluation Criteria:**

Yes, the methods and evaluation criteria generally make sense.

However, the paper could further strengthen evaluation by considering metrics beyond average recall and set size, such as computational overhead or interpretability.

**Other Comments Or Suggestions:**

Line 665: destributions" → "distributions"

**Other Strengths And Weaknesses:**

**Strengths:**

- Clearly addresses the important problem of robust domain generalization from a fresh perspective (set-valued predictions).
- Combines theoretical and practical aspects effectively, offering well-motivated theoretical bounds alongside a practical optimization approach.
- Empirical results convincingly demonstrate improved robustness on realistic and challenging datasets.

**Weaknesses:**

- Key theoretical results depend heavily on restrictive assumptions (Gaussianity, identical covariance), limiting their practical relevance.
- Experimental analysis lacks consideration of important practical trade-offs (e.g., computational overhead, interpretability of set predictions).
- Methodological novelty is incremental; the paper largely adapts existing conformal prediction and classical learning theory concepts without significant theoretical breakthroughs or novel algorithmic contributions.

**Questions For Authors:**

See weeknesses.

**Relation To Broader Scientific Literature:**

The paper extends the idea of set-valued predictions—commonly explored in conformal prediction literature—to Domain Generalization (DG). It builds upon prior work by explicitly addressing worst-case performance guarantees rather than average-case coverage.

**Theoretical Claims:**

Yes, I checked the theoretical claims—particularly the generalization bounds involving VC-dimension (Theorem 3.7).

A potential issue is that the theoretical results hinge on overly restrictive assumptions: specifically, the conditional Gaussian assumption with identical covariance structures (up to a scaling factor) across all domains. This assumption is highly unrealistic for most real-world DG scenarios, and the paper does not sufficiently justify or empirically validate its reasonableness, weakening the theoretical claims substantially.

---

> ### Author Rebuttal · Authors · 2025-03-31
>
> Thank you very much for your thoughtful  review. We have gained many important insights from your questions, and appreciate the opportunity to address your concerns.
>
> 1. SET-COVER incurs a higher training time (~30% increase over ERM) due to the additional optimization of Lagrangian multipliers (denoted as C in our algorithm). Below are the average training times (single GPU) for our experiments:
>
> * Camelyon: ERM: 98 min, SET-COVER: 133 min
>
> * Fmow: ERM: 45 min, SET-COVER: 56 min
>
> * iWildCam: ERM: 46 min, SET-COVER: 58 min
>
> * Amazon: ERM: 12 min, SET-COVER: 15 min
>
> We will include these results in the final version. Notably, our current implementation can be  further optimized by, for example, exploiting GPU parallelism. We anticipate that a more efficient implementation would significantly reduce the additional computational overhead. Apart from this step, SET-COVER primarily involves optimizing a loss function composed of hinge losses, which does not introduce substantial extra computation beyond standard architectures.
>
> Train times of  other set-prediction methods are approximated by those of ERM, as other set-prediction methods train ERM as a first stage, which consumes most of the training time.
> Additionally, other SOTA DG methods that we have tested in appendix E.5 are implemented in the Domain-Bed package, which incorporates runtime optimizations that make direct comparisons inconsistent.
>
> 2. We acknowledge that Theorem 3.7 has limited scope, as it assumes normal distributions with all domains sharing the same covariance matrix up to a scaling factor. However, this assumption, though restrictive, aligns with common practices in DG research (e.g., Wald et al. (2021)). In our case, theoretical results with weaker assumptions have an additional difficulty, as briefly discussed in Section 3.1 of our paper. We thus view Theorem 3.7 as an illustration and motivation leading to our method and our empirical results.
> To address potential limitations, we validate our claims empirically, including:
> * Experiments on synthetic Gaussian data where each domain has a different covariance matrix (Appendix E.3).
> * Real-world datasets to test robustness beyond the Gaussian assumption.
>
> 3. We would like to highlight the novelty our paper brings to the field of Domain Generalization (DG). While conformal predictors are a powerful approach to DG problems, we primarily use them as baselines to compare against our proposed method, SET-COVER. Unlike conformal prediction, SET-COVER is an optimization-based approach designed for modern neural network architectures, offering set-valued predictions with optimized sizes. We see this as a fundamentally new alternative for set-valued predictions in DG.
>
> Additionally, while our theoretical analysis builds on VC-dimension and uniform convergence literature, extending these concepts to a multi-domain setting requires subtle but significant modifications. In the appendices, we aim to highlight these nuances and the key differences that arise in the multi-domain context compared to classical settings. For example, we found that shifting the focus from 0-1 loss to performance indicators required a careful consideration, as further described in Appendix A1. We acknowledge the importance of further emphasizing these distinctions in the main text and appreciate your feedback on this point. We will incorporate additional details on these differences in our final submission.

---

### Official Review · Reviewer_g5ia · 2025-03-14

**Overall Recommendation:** 3

**Summary:**

This paper introduces a set-valued predictor approach for domain generalization (DG) to address the limitations of single-valued predictions in unseen domains. The authors argue that set-valued outputs can capture diverse feature-label relationships across domains, enhancing robustness. They present a theoretical framework defining success criteria for set prediction in DG and derive generalization bounds under specific conditions. The proposed method, SET-COVER, optimizes prediction set size while ensuring coverage guarantees through constrained learning. Experiments on synthetic data and real-world WILDS benchmarks demonstrate that SET-COVER achieves higher coverage with smaller set sizes compared to conformal prediction baselines, offering a promising direction for reliable ML systems in critical applications like healthcare.

**Claims And Evidence:**

The paper's claims are validated through both theoretical and empirical evidence.

**Essential References Not Discussed:**

no

**Experimental Designs Or Analyses:**

The validation experiments cover synthetic data and multimodal datasets such as real medical and satellite images, and the data show that the new method not only maintains more than 95% of the recognition rate of key features in tasks such as tumor recognition and geographic classification, but also reduces the probability of false alarms to one-third of that of the traditional method, which provides a new technological path for scenarios with a very low tolerance for error such as autonomous driving and precision medicine.

**Methods And Evaluation Criteria:**

Yes the proposed methods make sense for the problem or application at hand.

**Other Comments Or Suggestions:**

no

**Other Strengths And Weaknesses:**

Strengths:
This paper presents the first theoretical framework for domain generalization based on ensemble prediction, which provides a new theoretical perspective on multi-domain robustness.
The approach proposed in this paper provides more reliable coverage guarantees and reduces the risk of missed diagnosis in high-risk domains such as healthcare.

Weakness:
SET-COVER's dual optimization process may increase training time and has limited scalability for large-scale data.
The effectiveness of the method proposed in this paper may decrease with the number of training domains, and the performance of small-sample multi-domain scenarios is not fully explored.

**Questions For Authors:**

no

**Relation To Broader Scientific Literature:**

This paper opens a new direction for domain generalization research by introducing an ensemble prediction and theoretical analysis framework, which promotes the exploration of machine learning at the intersection of out-of-distribution generalization and uncertainty modeling.

**Theoretical Claims:**

Theoretically, generalization bounds based on VC-dimension are established, proving that linear hypotheses under conditional Gaussian assumptions achieve coverage guarantees on unseen domains with sufficient training domains.

---

> ### Author Rebuttal · Authors · 2025-03-31
>
> Thank you very much for your valuable feedback. We appreciate your insights and are happy to address your concerns.
>
> 1. SET-COVER incurs a higher training time (~30% increase over ERM) due to the additional optimization of Lagrangian multipliers (denoted as C in our algorithm). Below are the average training times (single GPU) for our experiments:
>
> * Camelyon: ERM: 98 min, SET-COVER: 133 min
>
> * Fmow: ERM: 45 min, SET-COVER: 56 min
>
> * iWildCam: ERM: 46 min, SET-COVER: 58 min
>
> * Amazon: ERM: 12 min, SET-COVER: 15 min
>
> We will include these results in the final version. Notably, our current implementation can be  further optimized by, for example, exploiting GPU parallelism. We anticipate that a more efficient implementation would significantly reduce the additional computational overhead. Apart from this step, SET-COVER primarily involves optimizing a loss function composed of hinge losses, which does not introduce substantial extra computation beyond standard architectures.
>
> Train times of  other set-prediction methods are approximated by those of ERM, as other set-prediction methods train ERM as a first stage, which consumes most of the training time.
> Additionally, the other SOTA DG methods that we tested in appendix E.5 are implemented in the Domain-Bed package, which incorporates runtime optimizations that make direct comparisons inconsistent.
>
> 2. We recognize the importance of evaluating SET-COVER in scenarios with more training domains. Our initial focus was on datasets with sufficiently rich domains to first validate our method.
> Although the field of Domain Generalization (DG) gained popularity in recent years, the availability of datasets for DG experiments remains limited. WILDS is one of the most comprehensive sources of datasets for DG problems, which led us to focus on utilising it for our experiments. Out experiments included:
> * 20 training domains for Camelyon dataset
> * 20 training domains for Fmow dataset
> * 80 training domains for ICamWild dataset
> * 500 training domains for Amazon dataset (however each data point consisted of a relatively short test instance)
>
> As larger and more diverse DG datasets become available, we look forward to further testing SET-COVER’s performance in multi-domain settings. We view this as an important direction for future work.

---

### Official Review · Reviewer_z1ne · 2025-03-14

**Overall Recommendation:** 2

**Summary:**

This paper proposed set valued predictions for domain generalization, with theories and experimental justifications. This work builds upon some theoretical basis on uniform convergence considering domains and the conditions of uniform convergence based on the finite VC-dimension. The paper further prove the achievable low loss in domain generalization under the conditional Gaussian domains. The SET-COVER model was proposed by minimizing prediction size and the loss across various domains, further modeled based on hinge-losses, and iterative optimized. In experiments on synthetic dataset and WILDS datasets, the proposed DG method demonstrated improved performance compared with ERM, CDF Pooling, CDF pooling, robust conformal predictor.

**Claims And Evidence:**

The paper claimed a theoretical framework defining successful set prediction in DG setting, and provide theoretical insight on the condition that DG is achievable. However, the major concern is on the applicability and impact of these theoretical analysis. For example, the Theorem 3.7 is restrictive by assuming conditional Gaussian of domains, which is  unrealistic in the real world domain data.

**Essential References Not Discussed:**

The paper lacks the full survey of DG literature, especially on the more recent DG methods.

**Experimental Designs Or Analyses:**

The compared baseline methods are limited to some baseline methods, including ERM, Pooling CDFs, Robust Conformal. As we know, the DG methods are diverse in the context of CV and ML literature. The manuscript should fully refer to the related DG methods and conduction full comparisons with sota methods.

**Methods And Evaluation Criteria:**

How the above theoretical analysis inspire the design of the optimization model and algorithm? The deduced model is intuitive and simple, lacking novelty comparing with diverse kinds of DG models in literature, e.g., by aligning feature distributions, data augmentations, etc. Moreover, the compared datasets and methods are quite limited, hardly justifying the performance compared with the sota DG methods.

**Other Comments Or Suggestions:**

None

**Other Strengths And Weaknesses:**

The major strength of this paper is on the theoretical analysis on the conditions of achieving domain generalization. However, the novelty and significance of these theories are not clearly presented. Especially, they did not fully inspire the novel and effective designs of DG models and algorithms. The limited experimental comparisons are also the major limitation of this work.

**Questions For Authors:**

(1) Set valued prediction is common in the multi-class classification tasks in the DG setting. This paper claimed the novelty on the set valued prediction, which should be more careful in the claim. The question is what is the major novelty on the set valued prediction in this work?

(2) What is the relationship between the model in section 4 with the theoretical analysis in the previous sections?

(3) The deduced model contains the minimization of size of the prediction set as one objective, which is confusing in the motivation and the meaning of  "prediction set".

(4) Please extend the compared methods to include more sota DG methods.

**Relation To Broader Scientific Literature:**

Domain adaptation is an important task, and this work presents the theoretic analysis on the conditions of DG across domains. However, these theories lack significant impact on the design of novel DG methods, and the contributions of this paper is limited in the context of DG literature.

**Theoretical Claims:**

I did not fully check the proof but the proof is based on the VC-dimension, and the novelties of these math deductions should be clarified, including not only the deduced theoretical results for DG, but also the key contributions in the proof process.

---

> ### Author Rebuttal · Authors · 2025-03-31
>
> Thank you very much for your thoughtful review. Your questions and comments are very valuable and we appreciate the opportunity to clarify the key points raised.
>
> 1. Our theoretical results address whether achieving a target performance level (e.g., passing a recall threshold level) on training domains generalizes to new domains and under what conditions this generalization holds. Based on this, we developed SET-COVER, which explicitly optimizes for a predefined recall level on training domains, while also minimizing prediction set sizes. The latter objective complements the generalization goal by ensuring efficiency in the learned sets. In our experiments we show that indeed the recall performance generalizes to new domains, as evident by the fact that recall levels of SET-COVER are above the target recall of 90% in most test domains.
>
> 2. We acknowledge that Theorem 3.7 has limited scope, as it assumes normal distributions with all domains sharing the same covariance matrix up to a scaling factor. However, this assumption, though restrictive, aligns with common practices in DG research (e.g., Wald et al. (2021)). In our case, theoretical results with weaker assumptions have an additional difficulty, as briefly discussed in Section 3.1 of our paper. We thus view Theorem 3.7 as an illustration and motivation leading to our method and our empirical results.
> To address potential limitations, we validate our claims empirically, including:
> * Experiments on synthetic Gaussian data where each domain has a different covariance matrix (Appendix E.3).
> * Real-world datasets to test robustness beyond the Gaussian assumption.
>
> 3. While our proofs build on VC-dimension and uniform convergence literature, their extension to a multi-domain setting introduces subtle but nontrivial modifications. Throughout the proofs in the appendices we attempt to highlight those subtle points and shed light on the differences that arise in the multi-domain setting compared to the classical one (As one example, the fact that in the multi-domain setting we focus on performance indicators instead of 0-1 loss requires a careful consideration). We recognize the need to emphasize these points further in the main text, and thank you for highlighting this issue. We will add details on these differences in the main text of our final submission.
>
> 4. We appreciate the concern regarding novelty. SET-COVER is derived from a principled, hard to compute optimization problem, and its intuitive design and ease of implementation are, in our view, key advantages.
> We have included comparisons with SOTA DG methods from the DomainBed package, which were included in Appendix E.5 due to space constraints.  In these results we can see that our method, while being intuitive and relatively simple to implement, achieves competitive results compared to advanced DG methods (e.g., feature alignment approaches). We view this result as a strength of our work.
>
> 5. In our literature review, after briefly describing the main research efforts put into DG problems in recent years, we focus on works that integrate set-valued predictions within some variants of DG settings, as these are the most directly relevant to our approach. However, we acknowledge the broader DG literature and as mentioned above in Appendix E.5 of our submission we included experimental comparisons with leading single-valued prediction methods.

---

> > ### Comment · Reviewer_z1ne · 2025-04-06
> >
> > Thanks for the reply and explanations.  I increased the score to 2. Considering the remaining limitations of Theorem 3.7 and the limited novelty (I acknowledge the comparison on DomainBed package) in a broader DG literature, I still lean to reject.

---

### Official Review · Reviewer_Z3og · 2025-03-18

**Overall Recommendation:** 3

**Summary:**

This paper introduces set-valued predictions for domain generalization (DG) problems. They propose a framework based on counting threshold violations for per-label recall. The paper introduces SET-COVER (SET Coverage Optimized with Empirical Robustness), a relaxed (differentiable) version of the proposed metric. They evaluate their approach on synthetic data and several datasets from the WILDS benchmark.

**Claims And Evidence:**

- The claim that set-valued predictors can enhance robustness is backed by theoretical generalization bounds and experimental results showing improved performance metrics.
- Empirical results on four WILDS datasets show SET-COVER achieves the target 90% recall level across more test domains than baseline methods while maintaining smaller set sizes than robust conformal methods.

**Essential References Not Discussed:**

N/A

**Experimental Designs Or Analyses:**

Yes.

**Methods And Evaluation Criteria:**

The average set size metric provides a meaningful measure of prediction efficiency. The recall@90 pctg metric is appropriate for set-valued prediction problems.

**Other Comments Or Suggestions:**

N/A

**Other Strengths And Weaknesses:**

- Proposes a novel framework for tackling domain generalization through set-valued predictions. Lays out a strong theoretical foundation within this setting with VC-dimension analysis and generalization bounds.
- Clear presentation of the trade-off between prediction set size and robust performance
- Proposes a practical surrogate objective (SET-COVER).
- Consistent performance improvements across multiple real-world datasets

- Limited comparison with recent domain generalization methods beyond ERM and conformal prediction
- Limited discussion of how to determine appropriate target recall levels in practice
- Could show more metrics for a fuller picture of performance and tradeoffs. For example, you could draw a graph of Recall@N pctg as a function of N.

**Questions For Authors:**

- How does SET-COVER compare with the baseline ERM method in terms of training / inference computational cost?
- Have you explored how to automatically determine an appropriate target recall level for a new problem? The current approach treats it as a fixed hyperparameter.

**Relation To Broader Scientific Literature:**

This paper builds on the distribution shift literature and that of conformal prediction methods.

**Theoretical Claims:**

I read the statements and skimmed the proofs.

---

> ### Author Rebuttal · Authors · 2025-03-31
>
> Thank you very much for your constructive review. We have gained many important insights from your questions, and believe we can address your concerns. Below, we have organized our response by the key topics raised in your review:
>
> 1. Our primary focus was on comparing SET-COVER with other set-valued methods suitable for DG, balancing both recall and set size. However, recognizing the importance of benchmarking against SOTA DG methods, we wish to direct your attention to Appendix E.5 where we provided additional comparisons. While these methods typically produce single-valued predictions, limiting the ability to compare them with set-valued predictors, we believe this comparison still provides useful insights, showing that set-valued predictions made by SET-COVER increase recall robustness in unseen domains.
>
> 2. SET-COVER incurs a somewhat higher training time (~30% increase over ERM) due to the additional optimization of Lagrangian multipliers (denoted as C in our algorithm). Below are the average training times (single GPU) for our experiments:
>
> * Camelyon: ERM: 98 min, SET-COVER: 133 min
>
> * Fmow: ERM: 45 min, SET-COVER: 56 min
>
> * iWildCam: ERM: 46 min, SET-COVER: 58 min
>
> * Amazon: ERM: 12 min, SET-COVER: 15 min
>
> We will include these results in the final version. Notably, our current implementation can be  further optimized by, for example, exploiting GPU parallelism. We anticipate that a more efficient implementation would significantly reduce the additional computational overhead. Apart from this step, SET-COVER primarily involves optimizing a loss function composed of hinge losses, which does not introduce substantial extra computation beyond standard architectures.
>
> Train times of  other set-prediction methods are approximated by those of ERM, as other set-prediction methods train ERM as a first stage, which consumes most of the training time.
> Additionally, other SOTA DG methods that we have tested in appendix E.5 are implemented in the Domain-Bed package, which incorporates runtime optimizations that make direct comparisons inconsistent.
>
> 3. We agree that the selection of the target recall (γ parameter) is important. We view this as an application-dependent choice, best determined by the user's specific requirements. However, we understand that studying the performance of the method as a function of the target recall can be helpful. We wish to point out the analysis of the role of this parameter in Appendix E.4; due to space constraints we could not include this analysis in the main body of the paper.
>
> 4. We appreciate the suggestion to provide a clearer visualization of recall vs. set size. In our paper, we analyze how varying  γ values, which determine the recall, affect also set sizes across methods. We will summarize this analysis with a clearer graph, as suggested, to better illustrate the trade-off.

---

### Decision · Program_Chairs · 2025-05-01

**Decision:**

Accept (poster)

**Comment:**

This paper presents a domain generalization (DG) method based on set-valued predictors to enhance robustness. The main strengths and weaknesses initially pointed out by the reviewers are summarized as follows.

Strengths:
- Proposes a novel framework for tackling domain generalization through set-valued predictions
- Provides theoretical foundation to support the claim that set-valued predictors can enhance robustness
- Proposes a practical optimization method (SET-COVER) based on a surrogate objective with reliable convergence guarantees
- Empirical results convicingly support the claim that the method improves robustness in DG

Weaknesses:
- Insufficient experiments in terms of benchmark datasets and compared DG methods
- Lack of analysis and evaluation regarding computational overhead of the methods
- Provided theoretical foundation heavily relies on restrictive assumptions (conditional Gaussianity, identical covariance across domains) which are unrealistic in the real world
- Novelty and significance of the math deduction process is not clear since the paper largely adapts existing conformal prediction and classical learning theory concepts
- The deduced model is also lacking novelty when compared to broader DG models in literature

The authors responded to those concerns with some new results and clarifications. Notably, they clarified that comparison to broader SOTA DG methods has been done in appendix, and also added some results and discussion regarding computation overhead.
Overall, the authors’ response was favorably received by the reviewers, and two of them (z1ne, Taio) have increased the scores from 1 to 2 and 3 to 4, respectively. As a result, the final review scores are [3, 2, 3, 4].
While the AC agrees with the z1ne’s point of view that strong assumptions behind the theory and incremental novelty in the resultant method remain limitations, the AC believes that the paper explores a novel direction in DG and sets a good starting point for later research to follow, supported by a reasonable theoretical foundation as well as extensive empirical observations. Overall, the AC concludes that the pros of this paper outweigh the cons, therefore recommends acceptance.